# GeoLoRA: Geometric integration for parameter efficient fine-tuning

**Steffen Schotthöfer**[1], **Emanuele Zangrando**[2], **Gianluca Ceruti**[3], **Francesco Tudisco**[2,4,5], **Jonas Kusch**[6]

[1]Computer Science and Mathematics Division, Oak Ridge National Laboratory, USA
[2]Gran Sasso Science Institute, L'Aquila, Italy
[3]Department of Mathematics, University of Innsbruck, Austria
[4]School of Mathematics and Maxwell Institute, University of Edinburgh, UK
[5]Miniml.AI Ltd, UK
[6]Department of Data Science, Norwegian University of Life Sciences, Norway
Link to source code

## Abstract

Low-Rank Adaptation (LoRA) has become a widely used method for parameter-efficient fine-tuning of large-scale, pre-trained neural networks. However, LoRA and its extensions face several challenges, including the need for rank adaptivity, robustness, and computational efficiency during the fine-tuning process. We introduce GeoLoRA, a novel approach that addresses these limitations by leveraging dynamical low-rank approximation theory. GeoLoRA requires only a single back-propagation pass over the small-rank adapters, significantly reducing computational cost as compared to similar dynamical low-rank training methods and making it faster than popular baselines such as AdaLoRA. This allows GeoLoRA to efficiently adapt the allocated parameter budget across the model, achieving smaller low-rank adapters compared to heuristic methods like AdaLoRA and LoRA, while maintaining critical convergence, descent, and error-bound theoretical guarantees. The resulting method is not only more efficient but also more robust to varying hyperparameter settings. We demonstrate the effectiveness of GeoLoRA on several state-of-the-art benchmarks, showing that it outperforms existing methods in both accuracy and computational efficiency.

## 1 Introduction

Large-scale pre-trained and fine-tuned models have significantly advanced the performance of deep learning models in assisting various natural language processing and computer vision tasks. However, their deployment often incurs substantial computational and memory costs due to the enormous number of trainable parameters. To address this, parameter-efficient fine-tuning (PEFT) methods have been developed, which modify a subset of model parameters while keeping the rest frozen. Among these, low-rank adaptation (LoRA) (Hu et al., 2021) has emerged as a prominent approach, allowing efficient fine-tuning by injecting low-rank updates into pre-trained model weights. Despite its efficiency, LoRA faces limitations in adaptively distributing the parameter budget across weight matrices, and its performance is sensitive to the choice of hyperparameters (Zhang et al., 2023).

Recent works, such as AdaLoRA (Zhang et al., 2023), DyLoRA (Valipour et al., 2023), and ReLoRA (Lialin et al., 2023), have attempted to improve LoRA by dynamically adjusting the rank of the low-rank adapters during training. While these methods enhance parameter efficiency, they are constructed as simultaneous descent methods and therefore do not guarantee convergence to optimal low-rank adapters. Methods that guarantee convergence to optimal adapters exist (Schotthöfer et al., 2022; Schotthöfer & Laiu, 2024; Zangrando et al., 2024). However, these require several gradient tapes per iteration and, therefore, have an intrinsically higher run time per training step.

In this paper, we introduce GeoLoRA (Geometric Low-Rank Adaptation), a novel dynamical low-rank training method for parameter-efficient fine-tuning. GeoLoRA leverages the dynamical low-rank approximation theory from matrix differential equations (Koch & Lubich, 2007b; Ceruti et al., 2022; 2023) and exploits the intrinsic low-rank geometry of the weight matrices to allocate the

parameter budget across the model adaptively. This dynamic allocation is facilitated by a novel training strategy that updates the low-rank factors in parallel, contrasting with other recent methods based on dynamical low-rank approximation theory (Schotthöfer et al., 2022; Schotthöfer & Laiu, 2024; Zangrando et al., 2024), which require individual gradient tapes computed sequentially per each low-rank factor. Instead, GeoLoRA requires a single backprop pass over the small-rank adapters, limiting its computational cost and making it faster than popular baselines such as AdaLoRA (Zhang et al., 2023). Moreover, GeoLoRA maintains the exact orthonormality of the low-rank factors, avoiding the ill-conditioning issues associated with well-known high-curvature challenges arising in low-rank optimization (Schotthöfer et al., 2022).

Through extensive experiments on the GLUE benchmark, Vision Transformers, and Stable Diffusion, we show that GeoLoRA outperforms existing PEFT methods both in terms of accuracy and computational efficiency.

Along with the experimental evaluation, we provide a thorough convergence analysis, showing convergence to stationary points under standard assumptions, and a detailed error-bound analysis, demonstrating that GeoLoRA's low-rank adaptation remains close to its full-rank counterpart throughout the training process. This robustness is critical in ensuring that the fine-tuning process does not diverge, even under challenging conditions.

Overall, the main contributions of this work are as follows:

- We show that standard common training methods for low-rank adapters do not necessarily reach a local optimum. (Section 3)
- We propose GeoLoRA, a dynamical low-rank training method for low-rank adapters that leverages low-rank geometry and matrix differential equations to achieve adaptive parameter allocation. (Section 4)
- GeoLoRA only requires a single gradient tape and one small-size SVD per training step, making it competitive with existing baselines such as AdaLoRA.
- We provide a convergence analysis and error bound guarantees for GeoLoRA, ensuring robust training behaviour and convergence to a stationary point. (Section 4.2)
- Extensive experimental results demonstrate the superior performance of GeoLoRA over existing methods, with improved accuracy and training speed. (Section 5)

## 2 RELATED WORK

The growing size of neural networks has led to significant computational and memory challenges during both training and deployment. Several strategies have been proposed to mitigate these issues, including sparsification (Guo et al., 2016; Molchanov et al., 2017; He et al., 2017) and quantization (Wu et al., 2016; Courbariaux et al., 2016). Among these, layer factorization has gained traction as an effective approach to reducing memory requirements. Layer factorization techniques have been applied successfully in both pre-training (Wang et al., 2021; Khodak et al., 2021; Schotthöfer et al., 2022; Schotthöfer & Laiu, 2024; Zangrando et al., 2024; Zhao et al., 2024) and fine-tuning scenarios (Hu et al., 2021; Valipour et al., 2023; Zhang et al., 2023; Hayou et al., 2024; Zhao et al., 2024; Lialin et al., 2023), demonstrating their versatility across various tasks.

Low-rank adapters such as LoRA (Hu et al., 2021) have become a standard approach for PEFT by applying low-rank corrections to pre-trained models. LoRA introduces a low-rank decomposition to the weight matrices of the model, significantly reducing the number of trainable parameters while preserving performance. Despite its efficiency, LoRA's effectiveness heavily relies on the selection of hyperparameters such as learning rates and parameter budgets (Zhang et al., 2023; Hayou et al., 2024). These limitations have spurred the development of rank-adaptive methods. AdaLoRA (Zhang et al., 2023) is a popular extension of LoRA, which dynamically adjusts the rank of the low-rank adapters during training. By incorporating an orthogonality regularizer and SVD-like adaptation, AdaLoRA aims to address the challenges of rank selection and adaptation. It outperforms static low-rank methods by automatically allocating parameter budgets based on the importance of each matrix component. DyLoRA (Valipour et al., 2023) provides an alternative approach that hierarchically adjusts the rank during training, demonstrating that higher-rank adapters can lead to better performance than very low-rank ones. DoRA (Mao et al., 2024) proposes to sample a set of rank-1 updates for each LoRA layer and to combine them into a rank-$r$ update. Optimal rank-1

components are chosen during fine-tuning using an importance score based on the norm of the LoRA layer.

Beyond fine-tuning, low-rank methods have been successfully applied during the training and pre-training phases of neural networks. Techniques such as Pufferfish (Wang et al., 2021), intrinsic dimension reduction (Aghajanyan et al., 2020), and DLRT (Schotthöfer et al., 2022) suggest that large deep learning models have an inherently low intrinsic dimensionality, making them amenable to low-rank approximations. These methods propose reducing the number of parameters during training, potentially improving both efficiency and generalization. Recent works in dynamical low-rank training have explored the use of geometric properties of the low-rank parameter space to improve training stability and convergence. For example, the geometry-aware training approach for tensor layers in Tucker format (Zangrando et al., 2024) dynamically adapts the rank of the factorized layers, ensuring robust convergence even when the initial rank estimation is inaccurate. This method leverages the Riemannian geometry of the parameter space to avoid the ill-conditioning commonly encountered in low-rank training. ReLoRA (Lialin et al., 2023) introduces a parameter-efficient training method by using multiple low-rank updates to effectively train high-rank networks. This method allows training larger models with significant memory savings and training speed improvements compared to conventional methods. GaLore (Zhao et al., 2024) introduces a memory-efficient training strategy by projecting gradients onto a low-rank subspace. This approach achieves significant memory savings while maintaining performance.

## 3 LOW-RANK OPTIMIZATION: WHAT CAN GO WRONG

This section aims to discuss the nature of the critical points and optimization trajectories obtained when using gradient-based strategies for low-rank parameters, and why a straightforward application of gradient-based steps to factorized adapters may lead to suboptimal results.

Consider a neural network layer of the form

$$\mathbf{z} = \sigma(W_{\mathrm{pt}}\mathbf{x} + USV^\top \mathbf{x}), \tag{1}$$

where $\sigma$ is an arbitrary activation function, $W_{\mathrm{pt}} \in \mathbb{R}^{n \times n}$ are the frozen pre-trained weights, and $U, V \in \mathbb{R}^{n \times r}$, $S \in \mathbb{R}^{r \times r}$ are the rank-$r$ adapter weights, with input $\mathbf{x}$. For simplicity, we omit the bias term. Low-rank adapters of the form $W = USV^\top \in \mathbb{R}^{n \times n}$ have gained popularity in recent approaches such as (Zhang et al., 2023), although our discussion extends to other equivalent formulations like $W = AB$ (Hu et al., 2021). The objective of the training process is to minimize a loss function $\mathcal{L}(W)$ to find an optimal adapter weight $W_\star$. For full-rank matrices ($r = n$), optimality requires that $\nabla_W \mathcal{L}(W_\star) = 0$. However, when $r < n$, this condition is generally unattainable due to the reduced parameter space. In this scenario, we seek a matrix $W_\star$ that is locally optimal within the low-rank parameter space, meaning no further reduction in the loss function $\mathcal{L}$ is possible in the neighborhood of $W_\star$. A necessary condition for local optimality can be expressed as $P(W_\star)\nabla\mathcal{L}(W_\star) = 0$, see e.g., (Sato, 2021, Theorem 3.4). For orthonormal $U$ and $V$, the projection operator $P(USV^\top)Z := UU^\top Z(I - VV^\top) + ZVV^\top$ represents the *orthonormal* projection of $Z$ onto the tangent space at $USV^\top$. If $W_\star$ is not a saddle point, then this condition ensures that no search direction within the tangent space of $W_\star$ can further decrease the loss. See also Appendix J. Note that this only guarantees local optimality, a limitation shared by all gradient-based optimizers.

Current training methods for low-rank adapters aim to optimize the low-rank factors with a single backpropagation pass to compute all the required gradients simultaneously. This boils down to integrating the following gradient flow equations for each individual factor

$$\begin{aligned}
\dot{U} &= -\nabla_U \mathcal{L} = -(\nabla_W \mathcal{L})VS^\top, \\
\dot{V} &= -\nabla_V \mathcal{L} = -(\nabla_W \mathcal{L})^\top US, \\
\dot{S} &= -\nabla_S \mathcal{L} = -U^\top \nabla_W \mathcal{L} V,
\end{aligned} \tag{2}$$

where we use the chain rule and the decomposition $W = USV^\top$ to derive the expressions for $\nabla_{U,S,V}\mathcal{L}$. Here, we have omitted the dependence on the time variable $t$, i.e., $U, S, V = U(t), S(t), V(t)$ for improved readability, and we use dots to denote time derivatives. An explicit time discretization with a time step size equal to the learning rate $\lambda$ leads to the simultaneous gradient descent updates commonly employed in conventional training methods for LoRA. At first glance, this

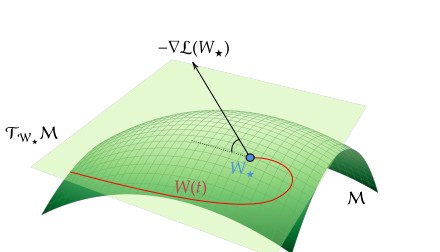 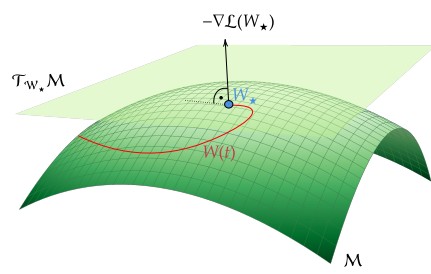

(a) Gradient flow of simultaneous gradient descent.   (b) Riemannian gradient flow.

Figure 1: Illustration of simultaneous vs. Riemannian gradient flow. The projector of the simultaneous gradient flow converges to a point $W_\star$ such that $\widehat{P}(W_\star)\nabla\mathcal{L} = 0$. Since $\widehat{P}$ is not an orthogonal projection, the gradient is not orthogonal to the tangent plane, i.e., $W_\star$ is suboptimal. For Riemannian gradient flows, the adapter converges to a point $W_\star$ such that $P(W_\star)\nabla\mathcal{L} = 0$. Since $P$ is the orthogonal projection on the tangent space, $W_\star$ is a local optimum, i.e., no directions exist in the tangent space $\mathcal{T}_{W_\star}\mathcal{M}$, which further decrease the loss. Here, $\mathcal{M}$ denotes the space of low-rank adapters, and $\mathcal{T}_{W_\star}\mathcal{M}$ represents the tangent space at the optimal adapter weight $W_\star$.

procedure appears effective, as a single update step will decrease the loss *if we freeze all but one of the low-rank factors*. However, in practice, LoRA training modifies all low-rank factors *simultaneously*, raising the question of how this affects the overall optimization trajectory.

To address this, consider the evolution equation for $W = USV^\top$, derived directly using the chain rule and eq. (2)

$$\dot{W} = \dot{U}SV^\top + U\dot{S}V^\top + US\dot{V}^\top$$
$$\overset{(2)}{=} -\nabla_W\mathcal{L}VS^\top SV^\top - UU^\top\nabla_W\mathcal{L}VV^\top - USS^\top U^\top\nabla_W\mathcal{L} =: -\widehat{P}(W)\nabla_W\mathcal{L}. \quad (3)$$

The operator $\widehat{P}(USV^\top)Z := ZVS^\top SV^\top + UU^\top ZVV^\top + USS^\top U^\top Z$ again represents a linear mapping onto the tangent space at $W = USV^\top$. Note that this projection depends on the individual low-rank factors $U$, $S$, and $V$, but we use the notation $\widehat{P}(W)$ for brevity. Simultaneous descent methods approximate the gradient flow of eq. (3), which ideally converges to a solution $W_\star$ such that $\widehat{P}(W_\star)\nabla\mathcal{L}(W_\star) = 0$. However, $\widehat{P}$ is orthogonal only when $U$ and $V$ are orthonormal and $S = I$, where $I$ denotes the identity matrix. If these conditions are not met, the resulting optimization process may not find an optimal weight within the low-rank parameter space. This is because $\widehat{P}(W_\star)\nabla\mathcal{L}(W_\star) = 0$ does not imply $P(W_\star)\nabla\mathcal{L}(W_\star) = 0$, thus there could still be some decrease direction along the tangent space as depicted in Figure 1.

To construct methods that converge to an optimal low-rank solution, an alternative approach is to evolve the adapter $W$ along the projected gradient flow $\dot{W}(t) = -P(W(t))\nabla\mathcal{L}(W(t))$. In this case, the corresponding evolution equations for the low-rank factors take the form

$$\dot{U} = -(I - UU^\top)\nabla_W\mathcal{L}VS^{-1},$$
$$\dot{V} = -(I - VV^\top)\nabla_W\mathcal{L}^\top US^{-\top}, \quad (4)$$
$$\dot{S} = -U^\top\nabla_W\mathcal{L}V,$$

assuming that $U$ and $V$ are orthonormal (Koch & Lubich, 2007b).

While the evolution defined in eq. (4) guarantees convergence to an optimal low-rank adapter, the presence of the $S^{-1}$ term on the right-hand side introduces stiffness in the gradient flow. This stiffness can significantly slow down convergence, especially when the singular values in $S$ vary greatly in magnitude. Robust solutions to address the stiffness problem and ensure convergence without being hindered by the $S^{-1}$ term have been proposed Schotthöfer et al. (2022); Zangrando et al. (2024); Schotthöfer & Laiu (2024). However, these methods require multiple gradient tape evaluations per training update, which makes them computationally more expensive than traditional LoRA training techniques with simultaneous updates.

To overcome these limitations, we propose GeoLoRA, a novel training method for low-rank adapters that only requires a single gradient tape evaluation per update while ensuring convergence to an optimal low-rank solution, following the projected gradient flow in eq. (4). This approach retains the computational efficiency of conventional LoRA methods while achieving comparable or even superior performance. By eliminating the need for multiple gradient tape evaluations, GeoLoRA offers a practical and scalable solution for training low-rank adapters effectively.

Before presenting the proposed training method, we illustrate different behaviours of different low-rank adaptation strategies using a toy example. Consider the problem of matching a rank-$r$ target matrix $W_{\text{target}} \in \mathbb{R}^{n \times n}$ with a low-rank adapter $W$, formulated as:

$$\min_W \frac{1}{2} \|W_{\text{target}} - W\|_F^2 \ . \tag{5}$$

We compare the convergence behavior of six different training methods for $n = 5000$, $r = 5$, and a learning rate of $\lambda = 0.1$:

1. Full fine-tuning (FT) (blue),
2. DLRT from (Schotthöfer et al., 2022) (orange),
3. The proposed GeoLoRA method (green),
4. Fixed rank LoRA from Hu et al. (2021) (red),
5. AdaLoRA from (Zhang et al., 2023) (brown),
6. Fixed rank AdaLoRA (purple).

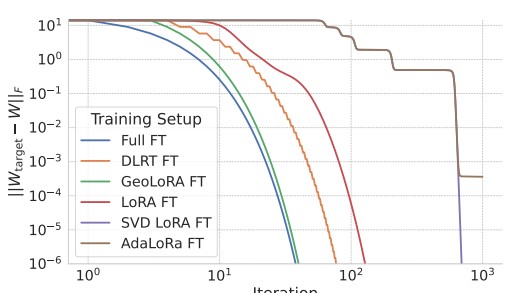

In this experiment, the fixed rank approaches (4, 6) use a rank of 50. All adapters $W$ are initialized to zero, with $S_0 = 0$ for the SVD-based methods (2, 3, 5, 6) and $B = 0$ for LoRA-based methods (4).

The results show that the proposed GeoLoRA method (3) converges as quickly as full fine-tuning. In contrast, method (2) (DLRT) takes approximately twice as long due to the sequential updates of the basis and coefficient matrices[1]. LoRA-type methods (4, 5, 6) exhibit slower convergence due to the suboptimality of the underlying gradient flow defined in eq. (2). AdaLoRA (5) solves the same gradient flow as method (6) but plateaus at a loss of $5 \times 10^{-4}$, corresponding to the regularization parameter for the terms $\|U^\top U - I\|_F^2 + \|V^\top V - I\|_F^2$ that enforce the orthonormality of $U$ and $V$. We had to fix the minimum rank to 5 for AdaLoRA to prevent stalling of the optimization due to rank underestimation, an issue not observed in methods (2) and (3), where rank augmentation avoided this problem.

## 4    THE PROPOSED METHOD

In this section, we introduce GeoLoRA (Geometric Low-Rank Adaptation) a novel low-rank fine-tuning method that integrates **rank adaptivity**, **low-rank optimality**, and **memory and computational efficiency**. Our method builds upon the parallel geometric low-rank integrator originally designed for model order reduction in high-dimensional PDEs (Ceruti et al., 2023), and it is equipped with loss descent, approximation bounds, and convergence guarantees. Notably, it improves upon existing dynamical low-rank methods, e.g. (Schotthöfer & Laiu, 2024; Zangrando et al., 2024; Schotthöfer et al., 2022) by updating basis and coefficients *in parallel* opposed to a *sequential* basis update and coefficient step. Moreover, only a single backward pass per iteration step is required through a novel evaluation strategy of robust gradients, thus doubling the wall-time performance. GeoLoRA is, therefore, the first low-rank training method solving the optimal gradient flow eq. (4) with training times per iteration comparable to standard simultaneous descent approaches to low-rank adaptation such as LoRA and AdaLoRA (Hu et al., 2021; Zhang et al., 2023). In particular, it improves upon these methods by providing robustness and convergence guarantees and demonstrating an overall improved performance and robustness to hyperparameters in numerical examples.

---

[1]The loss plateaus appear since the loss value remains constant during a basis update and only decreases during a coefficient update. This accounts for the fact that two gradient tapes need to be computed.

---

**Algorithm 1:** Single iteration of GeoLoRA.
The functions `optimizer_step`, `basis_augmentation`, and `truncation` are detailed in Algorithm 2 in the appendix.

---

**Input :** Initial orthonormal bases $U, V \in \mathbb{R}^{n \times r}$ and diagonal $S \in \mathbb{R}^{r \times r}$;
$\tau$: singular value threshold for rank truncation;
$\lambda$: learning rate.

1 Evaluate $\mathcal{L}(USV^\top)$                                    `/* Forward evaluate */`

2 $G_U \leftarrow \nabla_U \mathcal{L}(USV^\top); G_S \leftarrow \nabla_S \mathcal{L}(USV^\top); G_V \leftarrow \nabla_V \mathcal{L}(USV^\top)$     `/* Backprop */`

3 $\begin{cases} S^{\text{new}} \leftarrow \texttt{optimizer\_step}(S, G_S, \lambda) \\ K^{\text{new}} \leftarrow \texttt{optimizer\_step}(US, G_U S^{-\top}, \lambda) \\ L^{\text{new}} \leftarrow \texttt{optimizer\_step}(VS^\top, G_V S^{-1}, \lambda) \end{cases}$                  `/* in parallel */`

4 $\begin{cases} \widetilde{U} \leftarrow \texttt{basis\_augmentation}(U, K^{\text{new}}) \\ \widetilde{V} \leftarrow \texttt{basis\_augmentation}(V, L^{\text{new}}) \end{cases}$                  `/* in parallel */`

5 $\widehat{S} \leftarrow \begin{bmatrix} S^{\text{new}} & L^{\text{new},\top} \widetilde{V} \\ \widetilde{U}^\top K^{\text{new}} & 0 \end{bmatrix} \in \mathbb{R}^{2r \times 2r}$   `/* Assemble new coefficient matrix */`

6 $U, S, V, S^{-1} \leftarrow \texttt{truncation}(\widehat{S}, [U \mid \widetilde{U}], [V \mid \widetilde{V}])$

---

Starting from an initial factorization $U_0, V_0, S_0$ with initial rank $r_0$, where $S_0$ is diagonal and full-rank, GeoLoRA performs the following steps (also summarized in Algorithm 1):

1. **Perform a (stochastic) gradient step** to compute the new variables $S^{\text{new}} \in \mathbb{R}^{r_0 \times r_0}$, $L^{\text{new}} \in \mathbb{R}^{n \times r_0}$, and $K^{\text{new}} \in \mathbb{R}^{n \times r_0}$, as follows:

$$
\begin{aligned}
S^{\text{new}} &= S_0 - \lambda \nabla_S \mathcal{L}(U_0 S_0 V_0^\top) \\
K^{\text{new}} &= U_0 S_0 - \lambda \nabla_U \mathcal{L}(U_0 S_0 V_0^\top) S_0^{-\top} \\
L^{\text{new}} &= V_0 S_0^\top - \lambda \nabla_V \mathcal{L}(U_0 S_0 V_0^\top) S_0^{-1}.
\end{aligned}
\tag{6}
$$

We will see in Theorem 3 that using these variables mitigates the stiffness of the system in eq. (4) while approximating the optimal gradient flow. Note that the right-hand side gradients $\nabla_U \mathcal{L}, \nabla_V \mathcal{L}$, and $\nabla_S \mathcal{L}$ can be evaluated with only one backward pass through the network using standard algorithmic differentiation techniques, halving the computational cost of existing geometric methods such as (Schotthöfer et al., 2022; Zangrando et al., 2024). Evaluation of the inverse $S_0^{-1}$ induces no computational overhead since $S_0$ is diagonal at the start of each iteration.

2. **Augment the current bases** $U_0, V_0$ to twice their rank using the gradient dynamics of the loss, which is encoded in $K^{\text{new}}$ and $L^{\text{new}}$, i.e.

$$
\widehat{U} = [U_0, \widetilde{U}] = \text{ortho}([U_0, K^{\text{new}}]) \in \mathbb{R}^{n \times 2r_0} \quad \text{and} \quad \widehat{V} = [V_0, \widetilde{V}] = \text{ortho}([V_0, L^{\text{new}}]) \in \mathbb{R}^{n \times 2r_0}.
\tag{7}
$$

Here "ortho" denotes a column orthonormalization procedure such as the QR-algorithm. This augmentation step provides the low-rank adapter with a larger search space to increase the rank of its adaptation if the initial rank-guess $r_0$ was insufficient to fully capture the problem. Doubling the rank implies that in $\log(n)$ training iterations any rank can be captured by a rank one initialization, eliminating the need for tuning $r$ as a hyperparameter, see Figure 2.

3. **Assemble the augmented coefficient matrix**

$$
\widehat{S} \leftarrow \begin{bmatrix} S^{\text{new}} & L^{\text{new},\top} \widetilde{V} \\ \widetilde{U}^\top K^{\text{new}} & 0 \end{bmatrix} \in \mathbb{R}^{2r_0 \times 2r_0}
\tag{8}
$$

where we obtain the block entries $S^{\text{new}}, L^{\text{new}}$, and $K^{\text{new}}$ from eq. (6).

4. **Truncate redundant singular values** $s_i$ of $\widehat{S}$ and the corresponding singular vectors, i.e. basis functions of $\widehat{U}, \widehat{V}$, using the criterion

$$
\sum_{i=r_1+1}^{2r} s_i^2 < \vartheta,
\tag{9}
$$

where $r_1$ is the new rank of the factorization and $\vartheta$ is a tresholding hyperparameter. The singular values $s_i$ are obtained via the SVD of $\widehat{S} = P\Sigma Q^\top \in \mathbb{R}^{2r_0 \times 2r_0}$. Then we determine the new factorization as $S_1 = \text{diag}(s_1, \ldots, s_{r1}) \in \mathbb{R}^{r_1 \times r_1}$, $U_1 = \widehat{U} P_{(1,\ldots,r_1)} \in \mathbb{R}^{n \times r_1}$ and $V_1 = \widehat{V} Q_{(1,\ldots,r_1)} \in \mathbb{R}^{n \times r_1}$. The truncation threshold $\vartheta$ is chosen relative to the nuclear norm of the specific layer's current singular values, i.e. $\vartheta = \tau \|\widehat{S}\|_F^2$. Other norms, such as the 1-norm of the singular values $s_i$, are possible as well. Thus, the truncation threshold determines how aggressively to prune each layer individually. Analogously, the following global threshold similar to the one used in e.g. (Zhang et al., 2023; Ghadiri et al., 2023; Idelbayev & Carreira-Perpinan, 2020)

$$\sum_{\ell=1}^{L} \sum_{i=\ell+1}^{2r_\ell} s_{i,\ell}^2 < \frac{\tau}{1-\tau} \sum_{\ell=1}^{L} \sum_{i=1}^{r_{1,\ell}} s_{i,\ell}^2, \tag{10}$$

can be considered by summing the singular values across all the layers $\ell = 1, \ldots, L$. To directly control the parameter budget, order $s_{i,\ell}^2$ by descending by magnitude and selecting the largest ones first until either eq. (10) is violated or the budget is depleted.

### 4.1 PARAMETER INITIALIZATION

**LoRA-type adapters** (Hu et al., 2021) initilize the low rank matrices $B$, $A$ with zero initialization of $B$, and Gaussian initialization of $A$. This ensures that the fine-tuning indeed starts at the pretrained state of the network, i.e., $\sigma(W_{\text{pt}}\mathbf{x} + \frac{\alpha}{r}A_0 B_0^\top \mathbf{x}) = \sigma(W_{\text{pt}}\mathbf{x})$. For consistency with this initialization, the bases $U_0$ and $V_0$ can be initialized as random but orthonormal, whereas the coefficient matrix $S_0$ has zero-initialization. In this first solve of eq. (6), we set $S_0^{-1}$ as the identity matrix. As a result, the first solve of eq. (6) is inconsistent with the optimal dynamics of eq. (4). However, all following iterations evolve the low-rank trajectory according to the optimal gradient flow. Since in the first iterations of a LoRA fine-tuning, the adapter is typically close to the original solution but far from the fine-tuning optimum, this inconsistency is irrelevant to the overall convergence behavior of the method. Alternatively, the required gradients can be computed with three individual gradient tapes in the first iteration, which does not require the inversion of $S_0$.

The proposed method can readily be used for **dynamic low-rank compression** (Schotthöfer et al., 2022; Zangrando et al., 2024) of pre-trained networks, where we consider a layer $\mathbf{z} = \sigma(W\mathbf{x})$, and approximate $W \approx U_0 S_0 V_0^\top$. Here, the initial parameters $U_0, S_0, V_0$ are obtained by a truncated singular value composition of $W$. Finally, for **low-rank pre-training** of an untrained network with given architecture, i.e. predetermined layer dimensions $n$, but unknown rank $r$, the factors $U_0, V_0$ are initialized randomly, but orthonormal and $S_0$ is initialized randomly, but diagonal for easy initialization of $S_0^{-1}$.

### 4.2 ANALYSIS

In the following, we analyze Algorithm 1 under the general assumption that $\mathcal{L}$ is L-smooth with constant $L$ and bounded with constant $B$.

For brevity of exposition we denote $W_t^r = U_t S_t V_t^\top$ as the low-rank factorization at iteration $t$ evaluated with Algorithm 1, whereas $W_t$ denotes the full-rank solution obtained by "full fine-tuning" with stochastic gradient descent. Further, we denote by $f(W_t^r, \xi_t)$ the stochastic gradient of the network loss $\mathcal{L}$ w.r.t the low-rank weight $W_t^r$ at iteration $t$, obtained by batch-gradient descent. The i.i.d random variable $\xi_t$ models the randomness in the training data batch at iteration $t$. Lastly, recall that $P(W_t^r)Z$ denotes the orthogonal projection of the matrix $Z$ onto the tangent plane of the manifold of rank-$r$ matrices at the point $W_t^r$.

**Algorithm 1 is an optimizer on low-rank manifolds**: Theorem 1 shows, that the proposed scheme with stochastic gradients indeed decreases the training loss in each iteration, while optimizing on a manifold, and Theorem 2 yields stochastic convergence to a locally optimal stationary point.

**Theorem 1** (Stochastic descent estimate). *Algorithm 1 with stochastic (mini-batch) gradients fulfills*

$$\mathbb{E}_{\xi_{t+1}}[\mathcal{L}(W_{t+1}^r)] \leq \mathcal{L}(W_t^r) - \lambda\left(1 - \frac{L\lambda^2}{2}\right)\mathbb{E}_{\xi_t}[\|P(W_t^r)f(W_t^r, \xi_t)\|^2] + L\mathbb{E}_{\xi_t}[\|W_{t+1}^r - \widehat{W}_t^r\|]. \tag{11}$$

*where $W_t^r$, $\widehat{W}_t^r$, $W_{t+1}^r$ are the low-rank weight matrices at the start of iteration $t+1$, before, and after the truncation step, respectively.*

The proof is provided in Appendix D. The above theorem yields a loss descent guarantee up to the two last terms on the right-hand side. The first term of the right hand side induces the step size criterion $\lambda \leq \frac{2}{L}$, which resembles the step size criterion of full gradient descent, where the two right hand side terms read $-\lambda(1 - \frac{L\lambda}{2})\|f(W_t)\|^2$. This shows that the low-rank optimizer allows similar learning rates as a full fine-tuning setup, eliminating the need for the $\frac{\alpha}{r}$ scaling parameter of LoRA. The last term models the error introduced by the truncation step and is bounded by the user-determined cutoff threshold $\vartheta$, as $\mathbb{E}_{\xi_1}[\|W_{t+1}^r - \widehat{W_t^r}\|] \approx \vartheta$. As the solution stabilizes in rank, the error term vanishes, and we obtain the following main convergence result:

**Theorem 2** (Convergence). *Let $\mathcal{L} \geq 0$ and $W_1^r, \ldots, W_T^r$ be the solutions generated by Algorithm 1 over $T$ steps. Let the learning rate sequence $\{\lambda_t\}$ satisfy the Robbins-Monro conditions*

$$\sum_t \lambda_t = +\infty \qquad \sum_t \lambda_t^2 < +\infty \,,$$

*and each step $\lambda_t$ the step size restriction $\lambda_t \leq \frac{2}{L}$. Further assume $\sum_{t=1}^{T-1} \mathbb{E}[\|W_{t+1}^r - \widehat{W_t^r}\|] \leq D < \infty$, i.e. after some time, the solution $W_t^r$ is contained in a manifold of rank $r$. Then we have*

$$\liminf_{T \to \infty} \mathbb{E}[\|P(W_t^r)f(W_t^r)\|^2] = 0 \,,$$

*where the expected value is taken over all $\xi_t$.*

The proof is provided in Appendix E. Additionally, the solution trajectory of Algorithm 1 is close to the (full-rank) trajectory of the dynamical system

$$\dot{W}(t) = -\nabla_W \mathcal{L}(W(t)), \tag{12}$$

i.e., the gradient flow of full training or fine-tuning:

**Theorem 3** (Error-bound). *For an integer $k$, let $t = k\lambda$. Let $W(t)$ be the solution of eq. (12), and let $W_t^r$ be the factorized low-rank solution after $k$ steps with Algorithm 1. Assume that for any $Z$ in a neighborhood of $W(t)$, we have $\|(I - P(Z))\nabla\mathcal{L}(Z)\| < \varepsilon$, i.e., the gradient flow is close to $T_Z\mathcal{M}_r$. Then,*

$$\|W(t) - W_t^r\| \leq c_1\varepsilon + c_2\lambda + c_3\vartheta/\lambda \,. \tag{13}$$

*Moreover, let $W_{RF}(t)$ denote the solution of the Riemannian flow of eq. (4). Then,*

$$\|W_{RF}(t) - W_t^r\| \leq c_4\varepsilon + c_2\lambda + c_3\vartheta/\lambda \tag{14}$$

*where the constants $c_1$, $c_2$, $c_3$, $c_4$ depend only on $L$ and $B$.*

The proof is provided in Appendix G. We refer to Appendix F for an interpretation of Algorithm 1 as an integrator of the gradient flow of eq. (12).

Finally, we point out that the single-layer case discussed so far is not restrictive, and all the theoretical results above can be directly transferred to the multilayer setting by means of the following proposition:

**Proposition 1** (Global structure preservation). *The application of Algorithm 1 for multiple LoRA layers corresponds to the numerical integration of an augmented single matrix system on the adjacency matrix of the computational graph*

$$\dot{\mathcal{W}} = -P(\mathcal{W})\Pi\nabla\mathcal{L}(\mathcal{W})$$

*Where $\Pi$ is a linear projection that depends only on the structure of the neural network architecture. Moreover, the application of Algorithm 1 to this system, leads to the global truncation strategy proposed in Section 4.*

The proof of Proposition 1 can be found in Appendix I together with the relative derivation of the global truncation strategy.

## 5 NUMERICAL RESULTS

**DeBERTa for GLUE.** We evaluate the performance of GeoLoRA by fine-tuning the 183 million parameter transformer DeBERTaV3-base (He et al., 2023) on the GLUE Benchmark (Wang et al., 2019) and compare the results in Table 2. For details on the methods, implementation, hyperparameter choices, and benchmark setup, please refer to Appendix B.1. In most cases, GeoLoRA outperforms other methods on the benchmark, achieving better metrics with significantly fewer trainable parameters. This reduction in trainable parameters allows GeoLoRA to process substantially more samples during training and evaluation compared to AdaLoRA.

Table 1: Method comparison for low-rank finetuning. We compare the computational cost of a single training step for an $n \times n$ layer matrix of rank $r$. In the table, "local optimality" refers to the property $P(W_\star)\nabla\mathcal{L}(W_\star) = 0$ for the computed adapter $W_\star$, as discussed in Section 3.

| Method | compute (per iteration) | memory (per iteration) | # gradient evals. | rank adaptive | local optimality |
|---|---|---|---|---|---|
| Full FT | $\mathcal{O}(n^2)$ | $\mathcal{O}(n^2)$ | 1 | - | ✓ |
| GeoLoRA | $\mathcal{O}(2nr + (2n+1)r^2 + r^3)$ | $\mathcal{O}(4nr + 3r^2)$ | 1 | ✓ | ✓ |
| AdaLoRA (Zhang et al., 2023) | $\mathcal{O}(2nr + (2n+1)r^2 + r^3)$ | $\mathcal{O}(2nr + 3r^2)$ | 1 | ✓ | ✗ |
| DLRT (Schotthöfer et al., 2022) | $\mathcal{O}(6nr + (2n+5)r^2 + 9r^3)$ | $\mathcal{O}(4nr + 3r^2)$ | 3 | ✓ | ✓ |
| LoRA (Hu et al., 2021) | $\mathcal{O}(2nr)$ | $\mathcal{O}(2nr)$ | 1 | ✗ | ✗ |

Table 2: DeBERTaV3-base fine-tuning on GLUE. We compare with full fine-tuning (Full FT), Houlsby adapter (Houlsby et al., 2019) (HAdapter), Pfeiffer adapter (Pfeiffer et al., 2021) (PAdapter), LoRA (Hu et al., 2021), AdaLoRA (Zhang et al., 2023), DoRA (Mao et al., 2024), LoRA+(Hayou et al., 2024), and Bitfit(Zaken et al., 2022). We report target metrics and computational performance (higher is better) for the median of 5 runs using different random seeds. Best results per dataset are shown in bold. Results for BitFit, HAdapter, PAdapter were taken from (Zhang et al., 2023). "AdaLoRa matched" has the rank budget adpated to match the parameter count of GeoLoRA.

| Method (# Params) | MNLI (Acc) | SST-2 (Acc) | CoLA (Mcc) | QQP (F1) | QNLI (Acc) | RTE (Acc) | MRPC (Acc) | STS-B (Corr) | Mean |
|---|---|---|---|---|---|---|---|---|---|
| Full FT (184M) | 89.90 | 95.63 | 69.19 | 89.80 | 94.03 | 83.75 | 89.46 | 91.60 | 87.92 |
| BitFit (0.1M) | 89.37 | 94.84 | 66.96 | 84.95 | 92.24 | 78.70 | 87.75 | 91.35 | 85.77 |
| HAdapter (1.22M) | 90.13 | 95.53 | 68.64 | 89.27 | 94.11 | 84.48 | 89.95 | 91.48 | 87.94 |
| PAdapter (1.18M) | 90.33 | 95.61 | 68.77 | 89.40 | **94.29** | 85.20 | 89.46 | 91.54 | 88.07 |
| LoRA r=8 (1.33M) | 90.29 | 95.29 | 68.57 | 90.61 | 93.91 | 85.50 | 89.75 | 89.10 | 87.87 |
| LoRA+ r=8 (1.33M) | 90.31 | 95.37 | **69.22** | **90.82** | 93.96 | 85.50 | 89.55 | 88.07 | 87.85 |
| DoRA r=8 (1.33M) | 90.11 | 94.30 | 68.50 | 90.71 | 94.31 | 85.05 | 89.32 | 91.38 | 87.96 |
| AdaLoRA $r_{target} = 8$ (1.27M) | **90.44** | 95.64 | 68.76 | 90.65 | 94.11 | **86.00** | 89.44 | 91.41 | 88.30 |
| AdaLoRA, matched | 90.21 (0.75M) | 95.64 (1.27M) | 68.59 (1.07M) | 90.48 (0.72M) | 93.93 (0.72M) | 85.92 (1.16M) | 88.21 (0.74M) | 90.91(0.74M) | 88.28 (0.89M) |
| GeoLoRA | 90.38 (0.7M) | **95.98** (1.17M) | 69.03 (0.98M) | 90.53 (0.69M) | 94.23 (0.70M) | 85.93 (1.19M) | **90.10** (0.75M) | **91.58** (0.71M) | **88.47** (0.86M) |
| Evaluation and train time comparison | | | | | | | | | |
| AdaLoRA (eval/train) [it/sec] | 12.4/4.3 | 17.6/6.7 | 24.6/8.1 | 9.2/3.2 | 4.9/1.6 | 10.3/3.2 | 9.9/3.1 | 21.1/**8.5** | 13.75/4.83 |
| GeoLoRA (eval/train) [it/sec] | **17.1**/4.9 | **21.3**/**8.3** | **37.4**/**9.1** | **12.0**/**3.8** | **5.9**/**1.8** | **13.2**/**3.7** | **12.6**/**3.7** | **21.3**/8.3 | **17.6**/**5.6** |

**Performance analysis.** The proposed method from Algorithm 1 combines low-rank optimality guarantees with computational efficiency gains compared to existing low-rank optimization methods, as shown in Table 1. For a rank $r$ adapter, the computational cost of gradient evaluation (i.e., eq. (6)) is equivalent to that of AdaLoRA, which updates $U$, $S$, and $V$ directly, and is similar to a standard LoRA update. The cost of basis augmentation is $\mathcal{O}(nr^2)$ due to the QR decomposition in eq. (7), comparable to evaluating the orthonormality regularization terms in AdaLoRA. Rank truncation is performed via an SVD of $S$ at a cost of $\mathcal{O}(r^3)$, where typically $r \ll n$. The complexity analysis shows comparable per-iteration costs for LoRA, AdaLoRA, and GeoLoRA. In Table 2, we also report the number of iterations computed per second during training and evaluation for both GeoLoRA and AdaLoRA, demonstrating that GeoLoRA outperforms AdaLoRA across almost all GLUE benchmarks. We note that training and inference speed depend on both layer ranks and sequence lengths, and the performance difference is less pronounced for benchmarks with longer sequences.

**Vision transformer for object classification.** We compare GeoLoRA and AdaLoRA on fine-tuning the Vit-base-patch16-224 Vision Transformer, pre-trained on the Imagenet-1k dataset, and fine-tuned on Cifar10, Cifar100, and Tiny-Imagenet. GeoLoRA "local" uses a layer-wise rank truncation, and "global" uses the same global rank budget as AdaLoRA of 200 ranks. Details on implementation and hyperparameters are provided in Appendix B.2. Table 3 shows that GeoLoRA achieves higher validation accuracy than AdaLoRA, while using fewer trainable parameters.

**Ablations.** In Figure 2, we examine how the performance of GeoLoRA is influenced by the initial rank and learning rate. Figure 2(a, b) demonstrate that GeoLoRA dynamically recovers the intrinsic rank of the low-rank adaptation, regardless of the initial rank, highlighting the robustness of the method with respect to this hyperparameter. Notably, GeoLoRA can extend the adapter rank to full rank if necessary within logarithmic time, while truncating in constant time (in terms of optimization iterations). We provide a detailed discussion of the rank distribution across transformer layers in Appendix B.2. Similarly, Figure 2(c, d) show that GeoLoRA is less sensitive to learning rate variations compared to AdaLoRA.

**Dreambooth stable diffusion.** We test GeoLoRA on fine-tuning Stable Diffusion (Rombach et al., 2021) using Dreambooth (Ruiz et al., 2023) on their original datasets. Implementation details are

Table 3: Vit-base-patch16-224 fine-tuning on Cifar10, 100 and Tiny-Imagenet. We compare LoRa, AdaLoRA to GeoLoRA with local and global budgeting reporting the median of 5 runs using different random seeds. GeoLoRA "local" uses a layer-wise rank truncation, and "global" uses the same global rank budget as AdaLoRA.

| Method | Cifar 10 [%] | | Cifar 100 [%] | | Tiny-Imagenet [%] | |
|---|---|---|---|---|---|---|
| | # Params | Acc [%] | # Params | Acc [%] | # Params | Acc [%] |
| LoRA | 0.47M (r=3) | 98.47 | 0.47M (r=3) | 91.47 | 0.99M (r=6) | 87.34 |
| AdaLoRA | **0.47M** | 98.51 | 0.45M | 91.44 | 0.9M | 87.21 |
| GeoLoRA, local | **0.47M** | **98.55** | **0.35M** | **91.63** | 0.92M | **88.09** |
| GeoLoRA, global | 0.48M | 98.51 | 0.47M | 91.62 | **0.75M** | 88.07 |

Table 4: Stable Diffusion on Dreambooth benchmark. We compare LoRA and GeoLoRA reporting the median of 5 runs. For AdaLoRA, $r_0$ is the initial and $r$ is the target rank.

| Method | Val. Loss | # Params |
|---|---|---|
| LoRA ($r = 5$) | 0.275 | 3.0 M |
| LoRA ($r = 3$) | 0.281 | 1.8 M |
| AdaLoRA ($r_0 = 8, r = 5$) | 0.245 | 4.7M |
| AdaLoRA ($r_0 = 8, r = 3$) | 0.247 | 1.78M |
| GeoLoRA ($\tau = 0.02$) | **0.242** | 2.6M |
| GeoLoRA ($\tau = 0.1$) | 0.257 | **1.4M** |

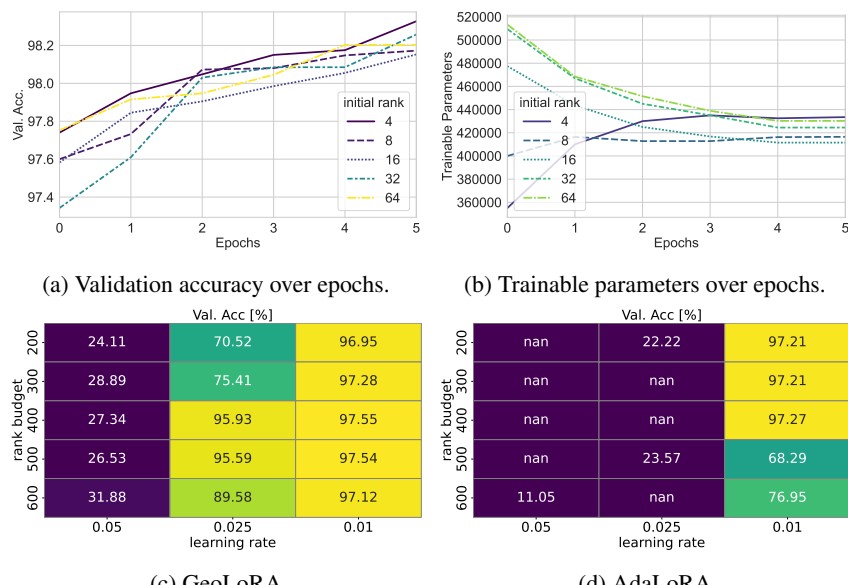

(a) Validation accuracy over epochs.

(b) Trainable parameters over epochs.

(c) GeoLoRA

(d) AdaLoRA

Figure 2: **Top panels (a, b):** GeoLoRA-adapted ViT-32b fine-tuned on Cifar10 with different initial layer ranks, using a learning rate of $1e-3$ and $\tau = 0.3$. The total number of trainable parameters converges to a similar steady state, regardless of the initial rank. The differences in validation accuracy between runs are smaller than the variance observed within individual setups. **Bottom panels (c, d):** GeoLoRA- and AdaLoRA-adapted ViT-32b fine-tuned on Cifar10 with different rank budgets and learning rates. Fields marked with `nan` indicate that training diverged within the first epoch. GeoLoRA demonstrates significantly greater robustness than AdaLoRA, particularly with high learning rates.

provided in Appendix B.5. As displayed in Table 4, GeoLoRA consistently achieves lower validation loss with fewer trainable parameters.

# 6 CONCLUSION

We introduced GeoLoRA (Geometric Low-Rank Adaptation), a novel adaptive low-rank fine-tuning method that combines computational efficiency with robustness. Based on geometric principles from dynamical low-rank approximation theory, the method comes with guarantees of convergence and local optimality. By leveraging a parallel update strategy of the low-rank adapters, the method requires only a single backward pass per iteration, achieving inference and training speed comparable or superior to existing baselines such as AdaLoRA, and much more efficient than previous geometric-aware strategies. Our experiments on the GLUE benchmark, Vision Transformers, and Stable Diffusion demonstrate that GeoLoRA outperforms existing PEFT methods in both accuracy and efficiency, with fewer trainable parameters. These results, alongside strong theoretical guarantees, position GeoLoRA as a robust solution for efficient model adaptation.

## FUNDING ACKNOLEDGEMENTS

This manuscript has been authored by UT-Battelle, LLC under Contract No. DE-AC05-00OR22725 with the U.S. Department of Energy. The United States Government retains and the publisher, by accepting the article for publication, acknowledges that the United States Government retains a non-exclusive, paid-up, irrevocable, world-wide license to publish or reproduce the published form of this manuscript, or allow others to do so, for United States Government purposes. The Department of Energy will provide public access to these results of federally sponsored research in accordance with the DOE Public Access Plan(`http://energy.gov/downloads/doe-public-access-plan`).

The work of E. Zangrando was funded by the MUR-PNRR project "Low-parametric machine learning". Francesco Tudisco is partially funded by the PRIN-MUR project MOLE code: 2022ZK5ME7 MUR D.D. financing decree n. 20428 of November 6th, 2024, CUP B53C24006410006; and by the PRIN-PNRR project FIN4GEO within the European Union's Next Generation EU framework, Mission 4, Component 2, CUP P2022BNB97.

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

---

**Algorithm 2:** Various auxiliary functions

---

1 **def** `optimizer_step`(*P: param, G: gradient, λ: learning rate*)**:**
2     $P^{\text{new}} \leftarrow P - \lambda G$        /* May use momentum and weight decay */
3     return $P^{\text{new}}$
4 **def** `basis_augmentation`(*B: old basis, $G_B$: basis dynamics*)**:**
5     $[B \mid \widetilde{B}] \leftarrow \texttt{qr}([B \mid G_B])$
6     return $\widetilde{B}$
7 **def** `truncation`(*$\widehat{S}$: augmented coefficient, $\widehat{U}$: augmented basis, $\widehat{V}$: augmented co-basis* )**:**
8     $P_{r_1}, \Sigma_{r_1}, Q_{r_1} \leftarrow$ truncated $\texttt{svd}(\widetilde{S})$ with threshold $\vartheta$ to new rank $r_1$
9     $U \leftarrow \widehat{U} P_{r_1}; V \leftarrow \widehat{V} Q_{r_1}$        /* Basis update */
10     $S \leftarrow \Sigma_{r_1}; S^{\text{inv}} \leftarrow \Sigma_{r_1}^{-1}$     /* Coefficient update with diagonal $\Sigma_{r_1}$ */
11     return $U, S, V, S^{\text{inv}}$

---

## A    ALGORITHM FOR AUXILIARY FUNCTIONS

We present the auxiliary function for Algorithm 1 in Algorithm 2.

## B    ADDITIONAL INFORMATION FOR THE NUMERICAL TEST CASES

### B.1    GLUE BENCHMARK

#### B.1.1    DATASET DESCRIPTION

We compare GeoLoRA to several fine-tuning methods from recent literature in the General Language Understanding Evaluation (GLUE) benchmark (Wang et al., 2019). The GLUE benchmark is a collection of diverse natural language understanding tasks designed to evaluate the performance of models in comprehending and processing human language. GLUE provides a comprehensive assessment by including tasks that cover a range of linguistic phenomena, such as textual entailment, sentiment analysis, sentence similarity, and more. The benchmark consists of nine different tasks:

- CoLA (Corpus of Linguistic Acceptability): Classifying whether a sentence is grammatically correct or not.
- SST-2 (Stanford Sentiment Treebank): Sentiment analysis task where the goal is to classify the sentiment of a sentence as positive or negative.
- MRPC (Microsoft Research Paraphrase Corpus): Identifying if two sentences are paraphrases of each other.
- STS-B (Semantic Textual Similarity Benchmark): Measuring the degree of semantic similarity between two sentences on a scale from 1 to 5.
- QQP (Quora Question Pairs): Determining if a pair of questions are semantically equivalent.
- MNLI (Multi-Genre Natural Language Inference): Classifying the relationship between a pair of sentences (entailment, contradiction, or neutral).
- QNLI (Question Natural Language Inference): Determining if a sentence provides a correct answer to a given question.
- RTE (Recognizing Textual Entailment): Binary classification task for entailment and contradiction.
- WNLI (Winograd Schema Challenge): Resolving pronoun reference ambiguity in sentences. Specific Focus: MRPC (Microsoft Research Paraphrase Corpus)

We present the benchmark overview in Table 5. To recapitulate, the F1 score is defined in dependence of precision score $P$ and recall score $R$. The model precision $P$ is given by

$$P := \frac{P_T}{P_T + P_F}, \tag{15}$$

Table 5: Summary of GLUE benchmark tasks

| Corpus | Task | #Train | #Dev | #Test | #Label | Metrics |
|--------|------|--------|------|-------|--------|---------|
| **Single-Sentence Classification (GLUE)** | | | | | | |
| CoLA | Acceptability | 8.5k | 1k | 1k | 2 | Matthews corr |
| SST | Sentiment | 67k | 872 | 1.8k | 2 | Accuracy |
| **Pairwise Text Classification (GLUE)** | | | | | | |
| MNLI | NLI | 393k | 20k | 20k | 3 | Accuracy |
| RTE | NLI | 2.5k | 276 | 3k | 2 | Accuracy |
| QQP | Paraphrase | 364k | 40k | 391k | 2 | F1 |
| MRPC | Paraphrase | 3.7k | 408 | 1.7k | 2 | Accuracy |
| QNLI | QA/NLI | 108k | 5.7k | 5.7k | 2 | Accuracy |
| **Text Similarity (GLUE)** | | | | | | |
| STS-B | Similarity | 7k | 1.5k | 1.4k | 1 | Pearson/Spearman cor |

where $P_T$ is the number of true positive and $P_F$ is the number of false positive examples. The recall $R$ is the ratio

$$R := \frac{P_T}{P_T + N_F},\tag{16}$$

where $N_F$ are the false negatives. The F1 score combines these two metrics to

$$F1 := \frac{2PR}{P + R}.\tag{17}$$

### B.1.2 REFERENCE IMPLEMENTATIONS

**Full finetuning (FT)**: This is the most common approach for model finetuning and transfer learning. Here, the model is initialized with pre-trained weights and all model parameters are updated with gradient descent.

**Bitfit (Zaken et al., 2022)**: Here, the model is initialized with pre-trained weights, but only bias terms are updated with gradient descent.

**Adapter tuning (Houlsby et al., 2019; Pfeiffer et al., 2021)**: Two-layer adapters are inserted between transformer blocks. In (Houlsby et al., 2019), the adapter is inserted between the self-attention module and the feed-forward module and equipped with a residual connection. In (Pfeiffer et al., 2021), the adapter is applied after the feed-forward module and the layer-norm module. To maintain conistency with the notation of (Zhang et al., 2023), we call the method of (Houlsby et al., 2019) HAdapter and the method of (Pfeiffer et al., 2021) PAdapter.

**LoRA (Hu et al., 2021)**: As stated in Section 3, LoRA applies additive corrections to selected weight matrices, i.e. $\mathbf{z} = \sigma(W_{\text{pt}}\mathbf{x} + \frac{\alpha}{r}AB^\top \mathbf{x})$ for $A, B \in \mathbb{R}^{n \times r}$. We apply LoRA to key $W_k$, query $W_q$ and value $W_v$ matrices of all attention blocks, and to both feed-forward layers $W_{f_1}$ and $W_{f_2}$. We chose the learning rates and optimizer as described in (Zhang et al., 2023), Appendix D-F.

The values in Table 1 for FT, Bitfit, Adapter tuning, and LoRA are taken from (Zhang et al., 2023). We compute the results for the methods DoRA, LoRA, LoRA+, and AdaLoRA using the HuggingFace open source implementations of the respective adpaters.

**DoRA (Mao et al., 2024)**: DoRA is an low-rank adapter similar to LoRA. The Main difference is that the $AB$ matrices are normalized and and additional magnitude parameter is included. Further, the adapter is initialized with the pretrained weights $W_0$ instead of zero initialization found in LORA.

**LoRA+ (Hayou et al., 2024)**: The key difference between standard LoRA and LoRA+ is in how learning rates are set. With standard LoRA, the learning rate is the same for $A$ and $B$. In LoRA+, different learning rates are set for $A$ and $B$, where the learning rate for $B$ is set as a multiple of that of $A$. The choice of the learning rates is the same of AdaLoRA (next paragraph), with a ratio $\lambda_B/\lambda_A = 1.1$.

**AdaLoRA (Zhang et al., 2023)**: As stated in Section 3, AdaLoRA applies additive corrections to selected weight matrices, i.e. $\mathbf{z} = \sigma(W_{\text{pt}}\mathbf{x} + \frac{\alpha}{r} USV^\top \mathbf{x})$ with arbitrary activation $\sigma$, frozen pre-trained weights $W_{\text{pt}} \in \mathbb{R}^{n \times n}$, rank $r$ adapter weights $U, V \in \mathbb{R}^{n \times r}$, $S \in \mathbb{R}^{r \times r}$. An SVD-based truncation mechanism is used to select layer ranks. Alternatively, the loss-sensitivity of singular vectors can be used for layer rank selection. Just like LoRA, we apply AdaLoRA to key $W_k$, query $W_q$ and value $W_v$ matrices of all attention blocks, and to both feed-forward layers $W_{f_1}$ and $W_{f_2}$.

We use the implementation of (Zhang et al., 2023, Appendix C) to compute the results for the presented reference methods and use the reported hyper-parameter choices of their Git Repository https://github.com/QingruZhang/AdaLoRA/tree/d10f5ebee16c478fa2f41a44a237b38e8c9b0338/NLU/scripts: We set the exponential moving average parameters $\beta_1$ and $\beta_2$ of AdamW as their default value 0.85. We select the learning rates as denoted in Table 6 and the regularization coefficient $\gamma$ as 0.1.

We compare against AdaLoRA, where we first match the total parameter budget to that of LoRA, i.e. choose the final budget $b^{(T)}$ of AdaLoRA as 576. Then we set $b^{(0)}$ as 1.5 times of $b^{(T)}$ In addition to the hyperparmeters chosen above, we compare AdaLoRA with budget levels obtained by GeoLoRA, where we again tune the final budget $b^{(T)}$ to approximately match the parameter count of GeoLoRA.

### B.1.3 IMPLEMENTATION DETAILS

We implement GeoLoRA as similar as possible as Adalora to achieve a fair comparison. That is, we add an adapter of the form $\mathbf{z} = \sigma(W_{\text{pt}}\mathbf{x} + USV^\top \mathbf{x})$ to the key $W_k$, query $W_q$ and value $W_v$ matrices of all attention blocks, and to both feed-forward layers $W_{f_1}$ and $W_{f_2}$. For each adapter, we employ Algorithm 1 to update the layer weights and ranks. All hyperparameters (except for the truncation tolerance $\tau$, which is unique to GeoLoRA) are identical to the hyperparameters in Lora+, DoRA, and AdaLoRA. The truncation tolerance $\tau$ is chosen as 0.15 across all datasets, i.e., singular values below this weighted threshold are set to zero. Note that all other low-rank adapters have similar hyperparameters to define the compression ratio specified above in the respective sections.

In Table 6, we display the hyper-parameter choices of GeoLoRA, Lora+, DoRA and AdaLoRA.

Table 6: Hyper-parameter setup for the GLUE benchmark. Learning rate, batch size, and number of epochs are adopted from the GitHub repository of AdaLoRA.

| Dataset | Learning Rate | Batch Size | # Epochs | $\tau$ (GeoLoRA) | inital rank (GeoLoRA) | $\frac{\lambda_B}{\lambda_A}$ (LoRA+) |
|---------|---------------|------------|----------|------------------|------------------------|----------------------------------------|
| MNLI | $5 \times 10^{-4}$ | 32 | 7 | 0.15 | 10 | 1.1 |
| RTE | $1.2 \times 10^{-3}$ | 32 | 50 | 0.15 | 10 | 1.1 |
| QNLI | $1.2 \times 10^{-3}$ | 32 | 5 | 0.15 | 10 | 1.1 |
| MRPC | $1 \times 10^{-3}$ | 32 | 30 | 0.15 | 10 | 1.1 |
| QQP | $5 \times 10^{-4}$ | 32 | 5 | 0.15 | 10 | 1.1 |
| SST-2 | $8 \times 10^{-4}$ | 32 | 24 | 0.15 | 10 | 1.1 |
| CoLA | $5 \times 10^{-4}$ | 32 | 25 | 0.15 | 10 | 1.1 |
| STS-B | $2.2 \times 10^{-3}$ | 32 | 25 | 0.15 | 10 | 1.1 |

### B.2 OBJECT CLASSIFICATION BENCHMARKS FOR THE VIT-BASE-PATCH16-224 VISION TRANSFORMER

We present in Table 3 results for finetuning the vit-base-patch16-224 vision transformer, which is pretrained on the imagenet-1k-dataset. The pretrained weights are downloaded from the torch-vision python package. For both AdaLora and GeoLoRA, we augment the key, query, and value matrices from attention layers as well as the three fully connected layers of each transformer block with a low-rank adapter. The biases of each layer are trainable. Additionally, the classifier is augmented with a low-rank adapter. The classifier is low-rank by construction, and we fix the rank as the number of classes. We fine-tune the vision transformer on Cifar10, Cifar100 and Tiny-Imagenet.

The hyperparameter settings to generate the results of Table 3, Figure 3 and Figure 4 are given in Table 7.

Figure 3 and Figure 4 show the rank distribution across layers for both AdaLoRA and GeoLoRA with global budget, for learning rate $\lambda = 1\mathrm{e}{-3}$ and $\lambda = 1\mathrm{e}{-4}$ and budgets ranging from $b = 200, \ldots, 600$

Table 7: Hyper-parameter setup for fine-tuning vit-base-patch16-224 vision transformer with Ge-oLoRA. AdaLoRA uses the same hyperparameters and the same rank budget for the global truncation as GeoLoRA.

| Dataset | Learning Rate | Batch Size | # Epochs | $\tau$ (local truncation) | rank budget (global truncation) | inital rank |
|---|---|---|---|---|---|---|
| Cifar10 | $1 \times 10^{-3}$ | 256 | 5 | 0.2 | 200 | 16 |
| Cifar100 | $1 \times 10^{-3}$ | 256 | 5 | 0.25 | 200 | 32 |
| TinyImageNet | $1 \times 10^{-4}$ | 256 | 5 | 0.15 | 300 | 32 |

total ranks for the network. Both methods prefer to allocate higher ranks to the deeper layers of the vision transformer, and prefer fully-connected layers over attention layers. Both methods pefer the first fully connected layer of a transformer block over the second. Overall GeoLoRA tends to assert higher ranks to single layers, compared to AdaLora, that distributes ranks more heterogeneously. The effects are more pronounced for smaller learning rates.

### B.3 ABLATION STUDY FOR THE INITIAL RANK FOR VIT

In addition to the results in Figure 2, we show in Table 8 that for Cifar10, Cifar100 and Tiny-Imagenet, GeoLoRA is robust with respect to the choice of the initial rank. We train using the hyperparameter of Table 7, but adapt the initial rank and fix the local truncation criterion $\tau = 0.15$.

Table 8: Vit-base-patch16-224 fine-tuning on Cifar10, 100 and Tiny-Imagenet. We report the median of 5 runs using different random seeds. GeoLoRA uses the layer-wise ("local") rank truncation with tolerance $\tau = 0.15$.

| Method | Cifar 10 [%] | | Cifar 100 [%] | | Tiny-Imagenet [%] | |
|---|---|---|---|---|---|---|
| | # Params | Acc [%] | # Params | Acc [%] | # Params | Acc [%] |
| GeoLoRA, local (r=10, $\tau = 0.15$) | 0.472M | 98.52 | 0.351M | 91.60 | 0.904M | 88.08 |
| GeoLoRA, local (r=16, $\tau = 0.15$) | 0.472M | 98.55 | 0.357M | 91.63 | 0.909M | 88.10 |
| GeoLoRA, local (r=32, $\tau = 0.15$) | 0.473M | 98.54 | 0.362M | 91.63 | 0.921M | 88.09 |

### B.4 LOSS CURVES FOR VIT

We consider the Cifar100 test case with the settings of Table 7, but adapt the learning rate to find the critical point, where GeoLoRA still converges well, but AdaLoRA diverges. A grid search between $5e-2$ and $1e-4$ for the learning rate yields $8e-3$. We display the corresponding training loss curves in Figure 5. For learning rates larger than $8e-3$, AdaLoRA diverges, whereas GeoLoRA remains stable for the entire range of learning rates. This observation agrees with the results of Figure 2, where we observe similar stability behavior in the example of Cifar10. We remark that AdaLoRA performs well for well-tuned learning rates, but the range of "good" learning rates is bigger for GeoLoRA in comparison with AdaLoRA.

### B.5 STABLE DIFFUSION ON DREAMBOOTH,

In this numerical example, we apply low-rank adapters to all linear and attention layers of the U-Net and the text encoder networks. The hyperparameters for LoRA and GeoLoRA are the same, apart from the fact that we start with adapters of rank 8 for both Unet and text encoder. We train for 5 full epochs, using adamW as an optimizer, with $(\beta_1, \beta_2) = (0.9, 0.999)$, initial learning rate $5 \times 10^{-6}$ and weight decay set to $10^{-2}$.

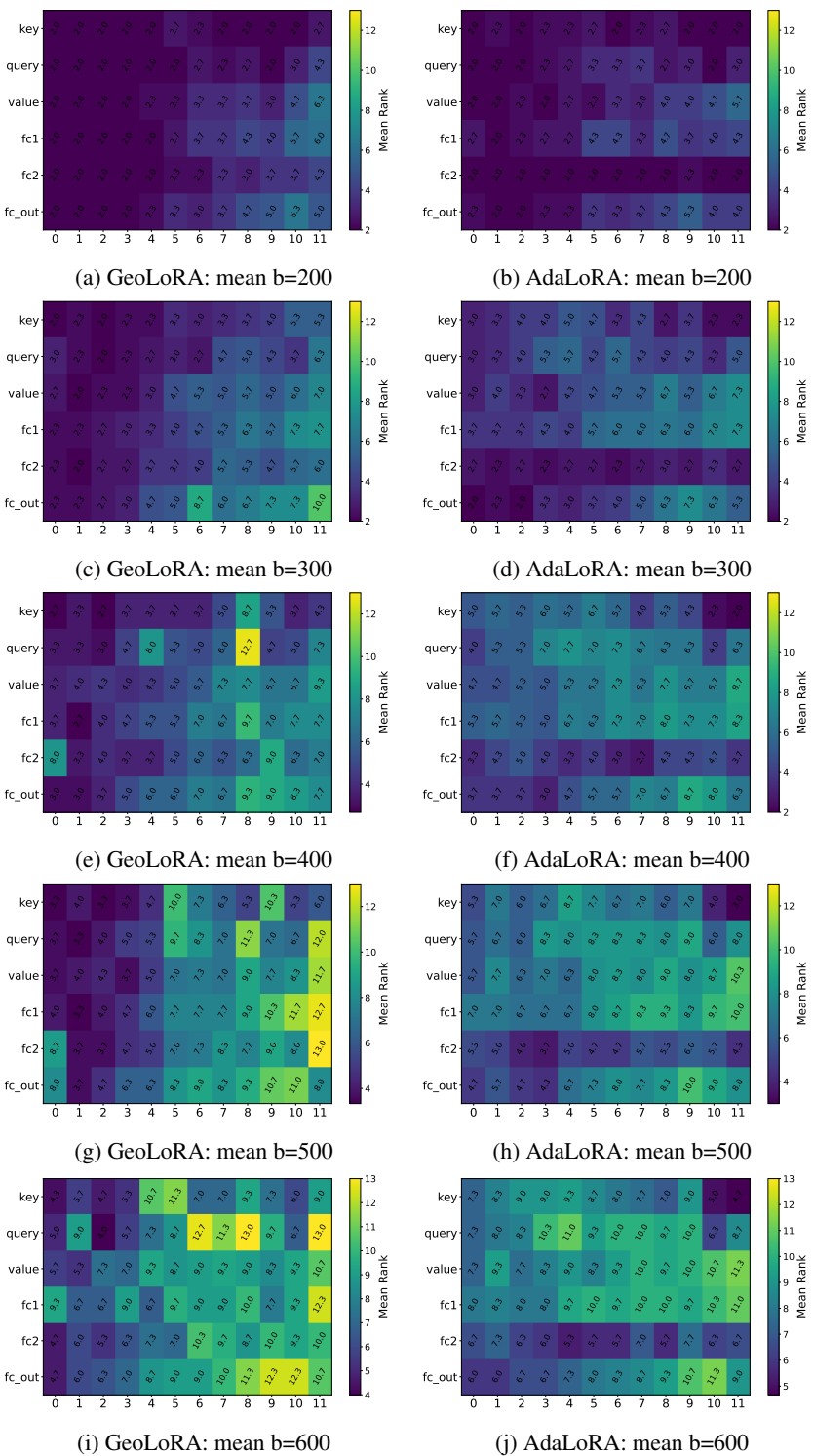

Figure 3: Rank distribution of Vit-32b finetuned on Cifar10 for 5 epochs at learning rate $1e-3$ using GeoLoRA and AdaLoRA.

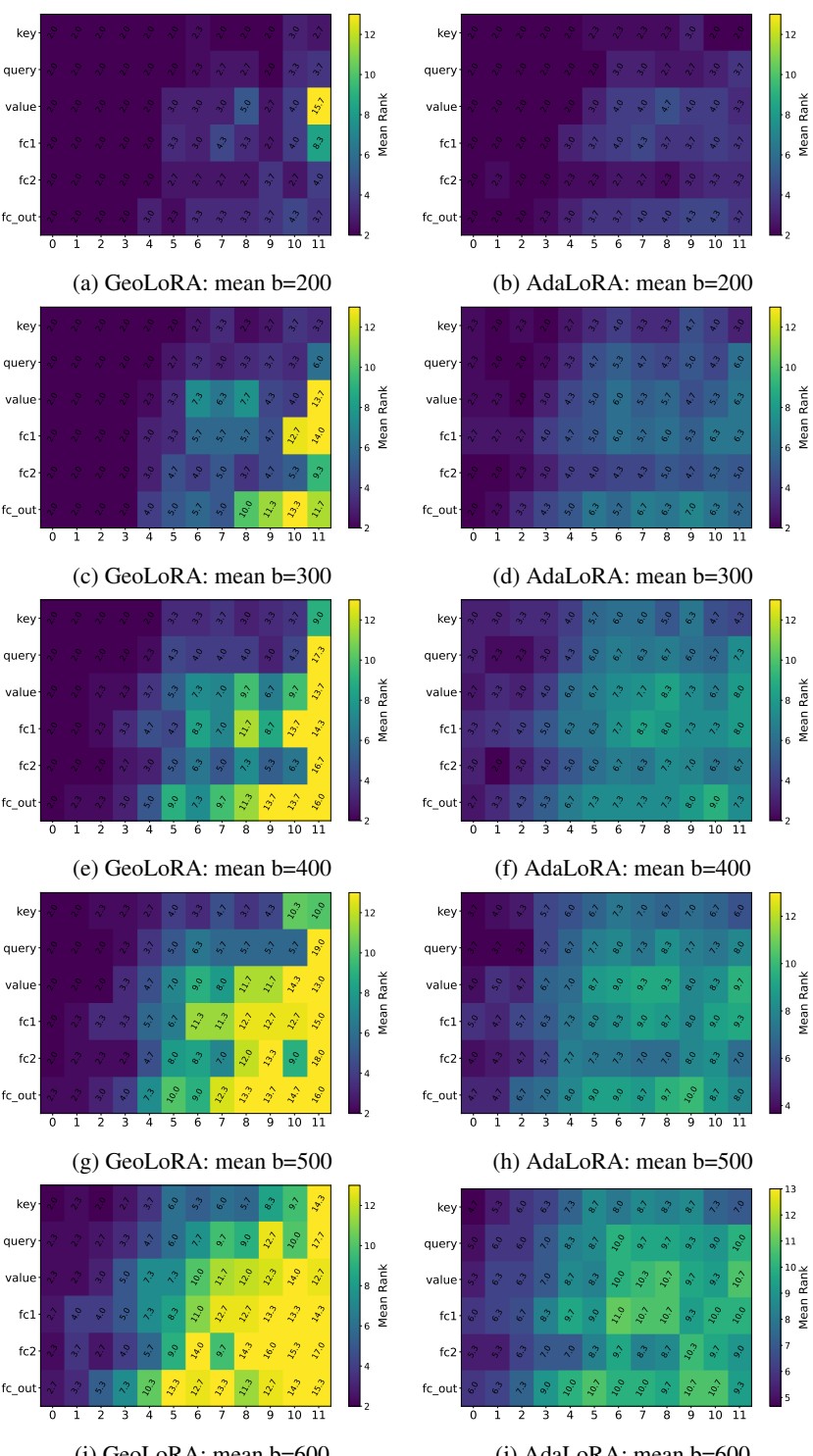

Figure 4: Rank distribution of Vit-32b finetuned on Cifar10 for 5 epochs at learning rate $1e-4$ using GeoLoRA and AdaLoRA.

# C OVERVIEW FOR THE NUMERICAL ANALYSIS

## C.1 NOTATION

We provide an overview of the notation used throughout the main manuscript and the appendix.

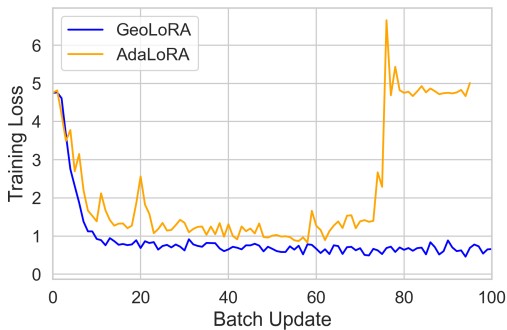

Figure 5: Loss evolution on a ViT trained on Cifar100.

- $W \in \mathbb{R}^{n \times n}$ is the full-rank weight matrix of a neural network layer or adapter.
- $Z \in \mathbb{R}^{n \times n}$ is an arbitrary matrix.
- $\mathcal{M}_r = \{Z \in \mathbb{R}^{n \times n} : \text{rank}(Z) = r\}$ is a manifold of rank $r$ matrices.
- $\mathcal{T}_Z \mathcal{M}_r$ is the tangent space of $\mathcal{M}_r$ at $Z$ for any $Z \in \mathbb{R}^{n \times n}$.
- $W^r = USV^\top \in \mathcal{M}_r$ is a rank-$r$ approximation of a matrix $W$.
- $\widehat{W}^r = \widehat{U}\widehat{S}\widehat{V}^\top \in \mathcal{M}_r$ is a rank-$r$ approximation of a matrix $W$ with augmented basis.
- $U, V \in \mathbb{R}^{n \times r}$ is the orthonormal basis and co-basis $\mathcal{M}_r$.
- $\widehat{U} = [U_0, \widetilde{U}] \in \mathbb{R}n \times 2r$ is the augmented basis. (Analogously for $\widehat{V}$.)
- $U_0 \in \mathbb{R}^{n \times r}$ is the basis at the beginning of the iteration. (Analogously for $V_0$.)
- $\widetilde{U} \in \mathbb{R}^{n \times r}$ is the basis augmentation, obtained by $[U_0, \widetilde{U}] = \text{ortho}([U_0, K^{\text{new}}]) \in \mathbb{R}^{n \times 2r_0}$. (Analogously for $\widetilde{V}$.)
- $S \in \mathbb{R}^{r \times r}$ is the coefficient matrix so assemble the low-rank approximation $W^r$ from $U, V$.
- $P(Z)$ is the orthogonal projection onto $\mathcal{T}_Z \mathcal{M}_r$.
- $P_U = UU^\top$ is the orthogonal projection onto the range of orthonormal $U \in \mathbb{R}^{n \times r}$.
- $P_V = VV^\top$ is the orthogonal projection onto the range of orthonormal $V \in \mathbb{R}^{n \times r}$.
- When applied to vectors, $\|\cdot\|$ denotes the Euclidean norm ($\ell_2$-norm). When applied to matrices, $\|\cdot\|$ denotes the Frobenius norm.
- $\mathcal{L}(W; \xi)$ denotes the loss function dependent on weight matrix $W$ and data sample randomness $\xi$. Commonly abbreviated by $\mathcal{L}(W)$.
- $f(W; \xi) = -\nabla_W \mathcal{L}(W; \xi)$ is the negative stochastic loss gradient w.r.t $W$. Commonly abbreviated by $f(W)$.
- $F(W) = \mathbb{E}_\xi[f(W, \xi)]$ the expectation of the random loss gradient, called the deterministic gradient.

## C.2    RECAP OF COMMONLY USED PROPERTIES

We recapitulate repeatedly used properties of the mathematical objects and notations introduced above.

- Per definition, we have for any $Z \in \mathbb{R}^{n \times n}$

$$P(W^r)Z = UU^\top Z + ZVV^\top - UU^\top ZVV^\top \tag{18}$$

- Since $\mathcal{T}_Z \mathcal{M}_r$ is a subspace of $\mathbb{R}^{n \times n}$ for any $Z \in \mathcal{M}_r$ we can decompose the gradients $F$ and $f$ into $F(Z) = M(Z) + R(Z)$ and $f(Z) = f(Z) + r(Z)$ for any $Z \in \mathcal{R}^{n \times n}$, where $M(Z), m(Z) \in \mathcal{T}_Z \mathcal{M}_r$.

## C.3    GLOBAL ASSUMPTIONS

The following provides a comprehensive overview of the global assumptions for made in the analysis section and proofs of the provided theorems. The assumptions are common in literature, see e.g. (Hnatiuk et al., 2024)

**Assumption 1.** *There is an $\varepsilon > 0$ such that $\|R(Z)\|, \|r(Z)\| \leq \epsilon$ for all $Z \in \mathcal{M}_r$*

**Assumption 2.** *$F$ and $f$ are bounded by a constant $B > 0$ and $L$-continuous w.r.t. $\|\cdot\|$ with constant $L > 0$.*

**Assumption 3.** *There is a constant $C > 0$ such that $\|F(Z) - f(Z)\| \leq C$.*

**Assumption 4.** *At initial time, we assume that the difference of a full-rank weight matrix $W_0$ and its low-rank counterpart $W_0^r$ is bounded by $\|W_0 - Y_0\| \leq \delta$ for $\delta > 0$.*

**Assumption 5.** *For all times, we have w.l.o.g $\mathcal{L}(t) > 0$.*

## D  DESCENT DIRECTION

We first state a few auxiliary lemmas, which provide common inequalities that will be used in the following analysis.

**Lemma 1.** *(Hnatiuk et al., 2024, Lemma 5.2) For any two matrices $Y_1, Y_2 \in \mathbb{R}^{n \times n}$ and an L-smooth $\mathcal{L}$ with constant $L$ it holds*

$$\mathcal{L}(Y_1) - \mathcal{L}(Y_2) \leq -\langle Y_1 - Y_2, f(Y_2)\rangle + \frac{L}{2}\|Y_1 - Y_2\|^2, \tag{19}$$

*where $f(Y) = -\nabla_Y \mathcal{L}(Y)$. Furthermore, it holds*

$$\mathcal{L}(Y_1) - \mathcal{L}(Y_2) \leq -\langle Y_1 - Y_2, F(Y_2)\rangle + \frac{L}{2}\|Y_1 - Y_2\|^2, \tag{20}$$

*where $F(Y) = -\mathbb{E}[\nabla_Y \mathcal{L}(Y)]$.*

The following results are primarily based on (Hnatiuk et al., 2024) and use the reformulation of truncated terms as proposed in (Zangrando et al., 2024). For ease of notation, we use $f(W_t^r) = f(W_t^r, \xi_t)$. Hence, randomness is not explicitly stated in our notation. Note that in this case, the factorized solution $W_1^r = U_1 S_1 V_1^\top$ is random since it depends on $f(W_1^r)$. When using expected values, we explicitly write down the corresponding random variable. That is, $\mathbb{E}_\xi[\cdot]$ is the expected value for a random variable $\xi$. We denote the random variable in step $T$ as $\xi_T$ and denote $\mathbb{E}[\cdot] := \mathbb{E}_{\xi_1, \cdots, \xi_T}[\cdot]$.

**Theorem 4.** *(Restatement of Theorem 1) Algorithm 1 with stochastic (mini-batch) gradients fulfills*

$$\mathbb{E}_{\xi_{t+1}}[\mathcal{L}(W_{t+1}^r)] \leq \mathcal{L}(W_t^r) - \lambda\left(1 - \frac{L\lambda^2}{2}\right)\mathbb{E}_{\xi_t}[\|P(W_t^r)f(W_t^r, \xi_t)\|^2] + L\mathbb{E}_{\xi_t}[\|W_{t+1}^r - \widehat{W}_t^r\|]. \tag{21}$$

*where $W_t^r$, $\widehat{W}_t^r$, $W_{t+1}^r$ are the low-rank weight matrices at the start of iteration $t + 1$, before, and after the truncation step, respectively. The step size is given by $\lambda$.*

*Proof.* Without loss of generality, we restrict ourselves to time steps $t = 0$ and write $f(W_0^r)$ shorthand for $f(W_{t=0}^r, \xi_t)$. By definition of the coefficient matrix assembly in eq. (8), we get respectively

- $\widetilde{U}\widetilde{U}^\top f(W_0^r)V_0 V_0^\top$ for the right hand side of the $S^{\text{new}}$ block

- $U_0 U_0^\top f(W_0^r)\widetilde{V}\widetilde{V}^\top$ for the right hand side of the $L^{\text{new}}$ block

- $\widetilde{U}\widetilde{U}^\top f(W_0^r)V_0 V_0^\top$ for the right hand side of the $K^{\text{new}}$ block

- and zero for the lower right block.

Since the augmented bases are orthonormal, we can write for $W_0^r = U_0 S_0 V_0$

$$\begin{aligned}
\widehat{W}_0^r &\overset{(8)}{=} W_0^r + \lambda U_0 U_0^\top f(W_0^r)V_0 V_0^\top + \lambda\widetilde{U}\widetilde{U}^\top f(W_0^r)V_0 V_0^\top + \lambda U_0 U_0^\top f(W_0^r)\widetilde{V}\widetilde{V}^\top \\
&= W_0^r - \lambda U_0 U_0^\top f(W_0^r)V_0 V_0^\top + \lambda\widehat{U}\widehat{U}^\top f(W_0^r)V_0 V_0^\top + \lambda U_0 U_0^\top f(W_0^r)\widehat{V}\widehat{V}^\top \\
&= W_0^r - \lambda U_0 U_0^\top f(W_0^r)V_0 V_0^\top + \lambda f(W_0^r)V_0 V_0^\top + \lambda U_0 U_0^\top f(W_0^r) \\
&\overset{(18)}{=} W_0^r + \lambda P(W_0^r)f(W_0^r).
\end{aligned}$$

By Lemma 1 we have

$$\mathcal{L}(\widehat{W}_0^r) - \mathcal{L}(W_0^r) \leq -\langle f(W_0^r), \widehat{W}_0^r - W_0^r \rangle + \frac{L}{2}\|\widehat{W}_0^r - W_0^r\|^2. \tag{22}$$

Therefore, plugging the above equation into eq. (22) yields

$$\mathcal{L}(\widehat{W}_0^r) - \mathcal{L}(W_0^r) \leq -\lambda\langle f(W_0^r), P(W_0^r)f(W_0^r)\rangle + \frac{L\lambda^2}{2}\|P(W_0^r)f(W_0^r)\|^2 \tag{23}$$

$$= -\lambda\langle P(W_0^r)f(W_0^r), P(W_0^r)f(W_0^r)\rangle + \frac{L\lambda^2}{2}\|P(W_0^r)f(W_0^r)\|^2 \tag{24}$$

$$= -\lambda\left(1 - \frac{L\lambda^2}{2}\right)\|P(W_0^r)f(W_0^r)\|^2. \tag{25}$$

where the second line is obtained by definition of the orthogonal projection. Comparing the loss before $\widehat{W}^r$ and after $W_1^r$ truncation yields for some $s \in (0,1)$ using the mean value theorem and the Cauchy-Schwarz inequality,

$$\mathcal{L}(W_1^r) \leq \mathcal{L}(\widehat{W}_0^r + \langle\nabla\mathcal{L}(sW_1^r + (1-s)\widehat{W}^r), W_1^r - \widehat{W}_0^r\rangle \leq \mathcal{L}(\widehat{W}_0^r) + L\|W_1^r - \widehat{W}_0^r\|. \tag{26}$$

Plugging eq. (26) into eq. (23) then gives

$$\mathcal{L}(W_1^r) - \mathcal{L}(W_0^r) \leq -\lambda\left(1 - \frac{L\lambda^2}{2}\right)\|P(W_0^r)f(W_0^r)\|^2 + L\|W_1^r - \widehat{W}_0^r\|,$$

where $L$ is the Lipschitz constant of $F$. Hence, taking the expected value yields

$$\mathbb{E}_{\xi_1}[\mathcal{L}(W_1^r)] \leq \mathcal{L}(W_0^r) - \lambda\left(1 - \frac{L\lambda^2}{2}\right)\mathbb{E}_{\xi_1}[\|P(W_0^r)f(W_0^r)\|^2] + L\mathbb{E}_{\xi_1}[\|W_1^r - \widehat{W}_0^r\|].$$

$\square$

# E  CONVERGENCE

**Theorem 5.** *(Restatement of Theorem 2) Let $\mathcal{L} \geq 0$ and $W_1^r, \ldots, W_T^r$ be the solutions generated by Algorithm 1 over $T$ steps. Let the learning rate sequence $\{\lambda_t\}$ satisfy the Robbins-Monro conditions:*

$$\sum_t \lambda_t = +\infty \qquad \sum_t \lambda_t^2 < +\infty.$$

*Further assume $\sum_{t=1}^{T-1} \mathbb{E}[\|W_{t+1}^r - \widehat{W}_t^r\|] \leq D < \infty$, i.e. after some time, the solution $W_t^r$ is contained in a manifold of rank $r$. Then we have*

$$\liminf_{T\to\infty} \mathbb{E}[\|P(W_T^r)f(W_T^r)\|^2] = 0,$$

*where the expected value is taken over all $\xi_t$.*

*Proof.* By taking the expected value over $\xi_1, \ldots, \xi_T$ in eq. (11) and denoting the corresponding expected value as $\mathbb{E}[\cdot]$ we get

$$\mathbb{E}[\mathcal{L}(W_{t+1}^r)] - \mathbb{E}[\mathcal{L}(W_t^r)] \leq -\lambda_t\mathbb{E}[\|P(W_t^r)f(W_t^r)\|^2] + \frac{L\lambda_t^2}{2}\mathbb{E}[\|P(W_t^r)f(W_t^r)\|^2]$$

$$+L\mathbb{E}[\|W_{t+1}^r - \widehat{W}_t^r\|]$$

$$= -\lambda_t\left(1 - \frac{L\lambda_t}{2}\right)\mathbb{E}[\|P(W_t^r)f(W_t^r)\|^2] + L\mathbb{E}[\|W_{t+1}^r - \widehat{W}_t^r\|].$$

Using a telescoping sum until $t = T$ then yields

$$-\mathcal{L}(Y_0) \leq \mathbb{E}[\mathcal{L}(W_t^r)] - \mathcal{L}(Y_0) \leq -\sum_{t=1}^{T-1}\lambda_t\left(1 - \frac{L\lambda_t}{2}\right)\mathbb{E}[\|P(W_t^r)f(W_t^r)\|^2]$$

$$+L\sum_{t=1}^{T-1}\mathbb{E}[\|W_{t+1}^r - \widehat{W}_t^r\|].$$

Rearranging gives

$$\sum_{t=1}^{T-1} \lambda_t \left(1 - \frac{L\lambda_t}{2}\right) \mathbb{E}[\|P(W_t^r)f(W_t^r)\|^2] \leq \mathcal{L}(Y_0) + L \sum_{t=1}^{T-1} \mathbb{E}[\|W_{t+1}^r - \widehat{W}_{t+1}^r\|].$$

$$\leq \mathcal{L}(Y_0) + LD.$$

Using the assumptions $\|P(W_t^r)f(W_t^r)\| \leq B$ and $\sum_{t=1}^{T-1} \mathbb{E}[\|W_{t+1}^r - \widehat{W}_{t+1}^r\|] \leq D$. Now, when $T \to \infty$, then the right-hand side remains bounded, implying that

$$\liminf_{T \to \infty} \mathbb{E}[\|P(W_t^r)f(W_t^r)\|^2] = 0.$$

$\square$

## F    EFFICIENT EVALUATION OF THE RIGHT HAND SIDE OF THE LOW-RANK DYNAMICS

Algorithm 1 creates a trajectory in the low-rank parameter space, that robustly follows the full-rank solution of the gradient flow of the neural network training. In particular, Theorem 6 yields a time-continuous representation of Algorithm 1.

**Theorem 6.** *The evolution equations eq. (6) are explicit Euler discretizations of a dynamical system which is equivalent to*

$$\begin{aligned}
\dot{S} &= -\nabla_S \mathcal{L}(U_0 S(t) V_0^\top), & S(t=0) &= S_0, \\
\dot{K} &= -\nabla_K \mathcal{L}(K(t) V_0^\top), & K(t=0) &= U_0 S_0, \\
\dot{L} &= -\nabla_L \mathcal{L}(U_0 L(t)^\top), & L(t=0) &= S_0^\top V_0,
\end{aligned} \tag{27}$$

*where $\mathcal{L}$ is the stochastic loss given random data samples.*

*Proof.* Consider the continuous time dynamics of $\dot{K}$, where we omit explicit time dependence on $U, S, V$ and $K$ for the sake of brevity, i.e.,

$$\begin{aligned}
\dot{K} &= (\dot{US}) \\
&= \dot{U}S + U\dot{S} \\
&\overset{(4)}{=} -(I - UU^\top)\nabla_W \mathcal{L}(USV^\top)VS^{-1}S - UU^\top \nabla_W \mathcal{L}(USV^\top)V \\
&= -(I - UU^\top)\nabla_W \mathcal{L}(USV^\top)V - UU^\top \nabla_W \mathcal{L}(USV^\top)V \\
&= (UU^\top - I)\nabla_W \mathcal{L}(USV^\top)V - UU^\top \nabla_W \mathcal{L}(USV^\top)V \\
&= -\nabla_W \mathcal{L}(USV^\top)V
\end{aligned} \tag{28}$$

Further, using the chain rule, we observe

$$\nabla_U \mathcal{L}(USV^\top) = \nabla_W \mathcal{L}(USV^\top)\nabla_U(USV^\top) = \nabla_W \mathcal{L}(USV^\top)VS^\top.$$

Thus, $-\nabla_U \mathcal{L}(USV^\top)S^{-\top} = -\nabla_W \mathcal{L}(USV^\top)V = \dot{K}$. Lastly we have by the chain rule $\dot{K} = -\nabla_W \mathcal{L}(USV^\top)V = -\nabla_K \mathcal{L}(USV^\top)$, which yields

$$\dot{K} = -\nabla_U \mathcal{L}(USV^\top)S^{-\top} = -\nabla_K \mathcal{L}(KV^\top).$$

Analogously we obtain for $\dot{L}$

$$\dot{L} = -\nabla_V \mathcal{L}(USV^\top)S^{-1} = -\nabla_L \mathcal{L}(UL^\top),$$

which concludes the proof. $\square$

Note that using an explicit Euler time discretization for eq. (27) directly yields eq. (6), the update step of GeoLoRA.

## G   Robust error bound of the low-rank system

We show the robust error bound for Algorithm 1 applied to a single layer, and then extend the result to a network containing multiple layers treated with Algorithm 1.

**Theorem 7.** *(Restatement of Theorem 3) For an integer $k$, let $t = k\lambda$. Let $W(t)$ be the solution of eq. (12), and let $W_t^r$ be the factorized low-rank solution after $k$ steps with Algorithm 1. Assume that for any $Z$ in a neighborhood of $W(t)$, we have $\|(I - P(Z))\nabla\mathcal{L}(Z)\| < \varepsilon$, i.e., the gradient flow is close to $T_Z\mathcal{M}_r$. Then,*

$$\|W(t) - W_t^r\| \le c_1\varepsilon + c_2\lambda + c_3\vartheta/\lambda. \tag{29}$$

*Moreover, let $W_{RF}(t)$ denote the solution of the Riemannian flow of equation 4. Then,*

$$\|W_{RF}(t) - W_t^r\| \le c_4\varepsilon + c_2\lambda + c_3\vartheta/\lambda \tag{30}$$

*where the constants $c_1$, $c_2$, $c_3$, $c_4$ depend only on $L$ and $B$.*

*Proof.* Let us first investigate the local error. That is, we choose the solution at a given time $t_0$ of the full-rank gradient flow of eq. (12), denoted as $W(t_0)$, as a given iteration of GeoLoRA, which we denote as $W_0^r$. Hence, $W(t_0) = W_0^r =: W_0 \in \mathcal{M}_r$. We are then interested in bounding the distance between the full-rank flow at $t_1 = t_0 + \lambda$ to the GeoLoRA solution after a single iteration with learning rate $\lambda$. To simplify notation, we denote $\widehat{U} = [U_0|\widetilde{U}] \in \mathbb{R}^{n \times 2r_0}$, $\widehat{V} = [V_0|\widetilde{V}] \in \mathbb{R}^{n \times 2r_0}$ and denote the projections onto these augmented basis vectors as $P_{\widehat{U}} = \widehat{U}\widehat{U}^\top$ and $P_{\widehat{V}} = \widehat{V}\widehat{V}^\top$. Moreover, $c$ denotes a generic constant that only depends on $L$ and $B$. It is important to note that this constant does not depend on $S_k^{-1}$, since we never perform Taylor expansions of the individual low-rank factors.

Let us denote the augmented solution of GeoLoRA before truncation as $\widehat{W}^r = \widehat{U}\widehat{S}\widehat{V}^\top$. Similarly, $W_1^r$ is the truncated solution after iteration 1. Then, the local error is bounded by

$$\|W(t_1) - W_1^r\| \le \|W(t_1) - P_{\widehat{U}}W(t_1)P_{\widehat{V}}\| +$$
$$\left\|P_{\widehat{U}}W(t_1)P_{\widehat{V}} - \widehat{W}^r\right\| + \left\|\widehat{W}^r - W_1^r\right\|.$$

In the following, we bound the three norms individually in three corresponding steps.

**Step 1** - Bounding $\left\|W(t_1) - P_{\widehat{U}}W(t_1)P_{\widehat{V}}\right\|$: Using the triangle inequality, we obtain

$$\left\|W(t_1) - P_{\widehat{U}}W(t_1)P_{\widehat{V}}\right\| \le \left\|W(t_1) - P_{\widehat{U}}W(t_1)\right\| + \left\|P_{\widehat{U}}W(t_1)(I - P_{\widehat{V}})\right\|$$
$$= \left\|(I - P_{\widehat{U}})W(t_1)\right\| + \left\|W(t_1)(I - P_{\widehat{V}})\right\|,$$

using orthonormality of $\widehat{U}$.

**First term:** Consider the first term with the dynamics $\dot{W}(t) = f(W)$ in mind,

$$\left\|(I - P_{\widehat{U}})W(t_1)\right\|$$
$$\stackrel{\text{(I)}}{\le} \left\|(I - P_{\widehat{U}})(W_0 + \lambda f(W_0))\right\| + c\lambda^2$$
$$\le \left\|(I - P_{\widehat{U}})(W_0 - \lambda P(W_0)f(W_0) + \lambda(I - P(W_0))f(W_0))\right\| + c\lambda^2$$
$$\le \left\|(I - P_{\widehat{U}})W_0\right\| + \lambda\left\|(I - P_{\widehat{U}})P(W_0)f(W_0)\right\| + \lambda\left\|(I - P_{\widehat{U}})(I - P(W_0))f(W_0)\right\| + c\lambda^2$$
$$\stackrel{\text{(II)}}{=} \lambda\left\|(I - P_{\widehat{U}})P(W_0)f(W_0)\right\| + \lambda\left\|(I - P_{\widehat{U}})(I - P(W_0))f(W_0)\right\| + c\lambda^2$$
$$\stackrel{\text{(III)}}{\le} \lambda\left\|(I - P_{\widehat{U}})P(W_0)f(W_0)\right\| + \lambda\varepsilon + c\lambda^2$$
$$\stackrel{\text{(IV)}}{\le} \lambda\left\|(I - P_{\widehat{U}})f(W_0)\widehat{V}\widehat{V}^\top\right\| + \lambda\varepsilon + c\lambda^2.$$

using Taylor expansion in (I), $W_0 \in \mathcal{M}_r$ in (II), Assumption 1 in (III), and eq. (18) in (IV).

By construction of the basis augmentation, we obtain

$$(I - P_{\widehat{U}})K^{\text{new}} = (I - P_{\widehat{U}})U_0S_0 = 0. \tag{31}$$

From eq. (31) we can directly conclude that $\left\|(I - P_{\widehat{U}})f(W_0)V_0 V_0^\top\right\| = 0$. Thus we obtain

$$\lambda\left\|(I - P_{\widehat{U}})f(W_0)\widehat{V}\widehat{V}^\top\right\| = \lambda\left\|(I - P_{\widehat{U}})f(W_0)V_0 V_0^\top\right\| + \lambda\left\|(I - P_{\widehat{U}})f(W_0)\widetilde{V}\widetilde{V}^\top\right\|$$
$$\leq \lambda\epsilon,$$

where we used for the second term that $\widetilde{V}$ is in the orthogonal complement of $V_0$. Hence,

$$\left\|(I - P_{\widehat{U}})W(t_1)\right\| \leq c\lambda^2 + \lambda\varepsilon.$$

**Second term**: The same derivation for the co-range using the evolution for $L(t)$ yields

$$\left\|W(t_1)(I - P_{\widehat{V}})\right\| \leq c\lambda^2 + \lambda\varepsilon.$$

**Step 2** - Bounding $\left\|P_{\widehat{U}}W(t_1)P_{\widehat{V}} - \widehat{W}^r\right\|$: We have by the assembly of the augmented $S$ matrix in eq. (8),

$$\widehat{W}^r = \widehat{U}\widehat{S}\widehat{V}^\top = U_0 S^{\text{new}}V_0^\top + \widetilde{U}\widetilde{U}^\top K^{\text{new}}V_0^\top + U_0 L^{\text{new},\top}\widetilde{V}\widetilde{V}^\top,$$

from which we obtain the error bound between the projected $W(t_1)$ and $\widehat{W}^r$:

$$\left\|P_{\widehat{U}}W(t_1)P_{\widehat{V}} - \widehat{W}^r\right\| \leq \left\|P_{\widehat{U}}W(t_1)P_{\widehat{V}} - U_0 S^{\text{new}}V_0^\top + \widetilde{U}\widetilde{U}^\top K^{\text{new}}V_0^\top + U_0 L^{\text{new},\top}\widetilde{V}\widetilde{V}^\top\right\|$$
$$\overset{(I)}{\leq} \left\|U_0^\top W(t_1)V_0 - S^{\text{new}}\right\| + \left\|\widetilde{U}^\top W(t_1)V_0 - \widetilde{U}^\top K^{\text{new}}\right\|$$
$$+ \left\|U_0^\top W(t_1)\widetilde{V} - L^{\text{new},\top}\widetilde{V}\right\| + \left\|\widetilde{U}^\top W(t_1)\widetilde{V}\right\|.$$

where we use orthonormality of $\widehat{U}, \widehat{V}$ in (I). All terms on the right-hand side can be bounded by $\lambda^2$ and $\varepsilon$ terms:

**First term:** We have

$$\left\|U_0^\top W(t_1)V_0 - S^{\text{new}}\right\| \overset{(I)}{=} \left\|\int_{t_0}^{t_1} U_0^\top(f(W(t)) - f(W_0))V_0\, dt\right\|$$
$$\overset{(II)}{\leq} \int_{t_0}^{t_1}\left\|f(W(t)) - f(W_0)\right\| dt$$
$$\overset{(III)}{=} \int_{t_0}^{t_1}\left\|f(W(t_0)) - f(W_0)\right\| dt + c\lambda^2$$
$$\overset{(IV)}{=} c\lambda^2$$

where we use in (I) $S^{\text{new}} = S_0 - U_0^\top \nabla_W \mathcal{L}(W_0;\xi)V_0 = -U_0^\top f(W_0)V_0$. We use the orthonormality of $U_0, V_0$ in (II), perform a Taylor expansion of the full-rank flow in (III), and finally use that $W(t_0) = W^S(t_0)$ in (IV).

**Second and third term:** We have

$$\left\|\widetilde{U}^\top W(t_1)V_0 - \widetilde{U}^\top K^{\text{new}}\right\| \overset{(I)}{\leq} \int_{t_0}^{t_1}\left\|\widetilde{U}^\top(f(W(t)) - f(W_0))V_0\right\| dt$$
$$\overset{(II)}{\leq} \int_{t_0}^{t_1}\left\|f(W(t_0)) - f(W_0)\right\| dt + c\lambda^2$$
$$= c\lambda^2,$$

where we use the K-step of GeoLoRA in (I) and a Taylor expansion of the full-rank flow in (II). $\left\|U_0^\top W(t_1)\widetilde{V} - L^{\text{new},\top}\widetilde{V}\right\|$ can be bounded analogously.

**Fourth term:** Lastly, we obtain for the fourth term,

$$
\begin{aligned}
\left\| \widetilde{U}^\top W(t_1)\widetilde{V} \right\| &= \left\| \widetilde{U}^\top W(t_0)\widetilde{V} + \int_{t_0}^{t_1} \widetilde{U}^\top f(W(t))\widetilde{V}\, dt \right\| \\
&\overset{(\mathrm{I})}{\leq} \int_{t_0}^{t_1} \left\| \widetilde{U}^\top f(W(t))\widetilde{V} \right\| dt \\
&\leq \int_{t_0}^{t_1} \left\| \widetilde{U}^\top f(W(t_0))\widetilde{V} \right\| dt + c\lambda^2 \overset{(\mathrm{II})}{\leq} \lambda\varepsilon + c\lambda^2 \, .
\end{aligned}
$$

with $\widetilde{U}^\top W(t_0)\widetilde{V} = 0$ by the construction of the augmented matrix $\widehat{S}$ used in (I), and in (II), we use Assumption 1.

**Step 3** - Bounding of $\left\| \widehat{W}^r - W_1^r \right\|$: By construction of the truncation step we directly obtain

$$
\left\| \widehat{W}^r - W_1^r \right\| \leq \vartheta
$$

In conclusion, we obtain for a single iteration of Algorithm 1

$$
\begin{aligned}
\| W(t_1) - W_1^r \| &\leq \left\| W(t_1) - P_{\widehat{U}} W(t_1) P_{\widehat{V}} \right\| + \\
&\quad \left\| \widehat{U}\widehat{U}^\top W(t_1) P_{\widehat{V}} - \widehat{W}^r \right\| + \left\| \widehat{W}^r - W_1^r \right\| \\
&\leq \widetilde{c}_1 \lambda\epsilon + \widetilde{c}_2 \lambda^2 + \vartheta
\end{aligned}
$$

To conclude, the global error in the training epochs follows by using the Lipschitz continuity of the gradient flow: We move from the local error in time to the global error in time by a standard ODEs argument of Lady Windermere's fan (Wanner & Hairer, 1996, §II.3); With $t = k\lambda$ and denoting the adapter computed with GeoLoRA at iteration $k$ as $W_t^r$ we then have

$$
\| W(t) - W_t^r \| \leq c_1 \epsilon + c_2 \lambda + c_3 \vartheta / \lambda \, .
$$

This bounds the distance between the full-rank flow and GeoLoRA. The result trivially extends to the Riemannian flow of equation 4. Denote by $W_{\mathrm{RF}}(t)$ the solution of the Riemannian flow $\dot{W}_{\mathrm{RF}}(t) = -P(W_{\mathrm{RF}}(t))\nabla_W \mathcal{L}(W_{\mathrm{RF}}(t))$. Then, since $\| W(t) - W_{\mathrm{RF}}(t) \| \leq c\epsilon$, it directly follows that

$$
\| W_{\mathrm{RF}}(t) - W_t^r \| \leq c_4 \epsilon + c_2 \lambda + c_3 \vartheta / \lambda \, .
$$

$\square$

## H    VISUALIZATION OF THE STIFFNESS OF THE BASIC LOW-RANK SYSTEM

Consider Equation (5) in the case for $n = 20$. We set the target matrix

$$
W = \begin{bmatrix} 0 & 15 & 0 & \ldots \\ -2 & 0 & 0 & \ldots \\ 0 & 0 & 0 & \ldots \\ \vdots & \vdots & \vdots & \ddots \end{bmatrix} \in \mathbb{R}^{20 \times 20},
$$

which has rank $r = 2$ and singular values $\sigma_1 = 15$ and $\sigma_2 = 2$ We compare SVD-lora, AdaLora, and GeoLoRA, both with an ansatz of form $W_{\mathrm{ans}} = USV^\top$ initialized as

$$
U, V = \begin{bmatrix} I \\ 0 \end{bmatrix} \in \mathbb{R}^{20 \times 4}, \qquad S = \begin{bmatrix} 10 & 0 & 0 & 0 \\ 0 & 1e-2 & 0 & 0 \\ 0 & 0 & 1e-4 & 0 \\ 0 & 0 & 0 & 1e-6 \end{bmatrix} \in \mathbb{R}^{4 \times 4}
$$

where $U, V$ are orthonormal, and the $S$ matrix has a fast decaying singular spectrum.

AdaLora and GeoLoRA use a relative singular value truncation threshold $\tau = 0.15$ for rank truncation. We found that learning rate $\lambda = 0.178$ is the maximal learning rate before AdaLora and SVD-Lora

become unstable, whereas GeoLoRA allows for arbitrary large learning rates, and we set $\lambda = 0.1$. We present the trajectories of the S-matrix elements of the corresponding methods in Figure 6 for up to $1000$ iterations or until single precision accuracy is reached. As seen in Figure 6, AdaLora and SVD-Lora exhibit heavy oscillations in the trajectories of the $S$-matrix elements - leading to slow convergence. Adalora - although using orthonormalization by regularization of the low-rank basis is not able to stabilize the training, leading to overestimation of the rank, which is $r = 5$ at final time and a final loss value of $1.6$. Similarly SVD-Lora exhibits even stronger oscillations and is not able to find the right matrix approximation. In contrast, GeoLoRA identifies the correct rank $r = 2$ and the corresponding correct singular values 15 and 2.

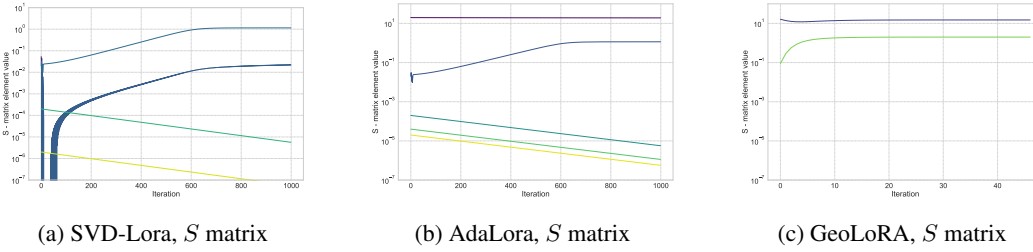

|  (a) SVD-Lora, $S$ matrix | (b) AdaLora, $S$ matrix | (c) GeoLoRA, $S$ matrix |

Figure 6: Time-trace of the matrix elements of SVD-Lora (a) AdaLora (b) and the proposed method GeoLoRA (c) to solve Equation (5). SVD-Lora was trained with learning rate $\lambda = 0.00178$, which is the largest learning rate for which the optimization remained stable, GeoLoRA allows larger learning rates, set to $\lambda = 0.1$. GeoLoRA converges fast to single precision accuracy, whereas SVD-LORA still has a loss value of 1.7 after 1000 iterations, due to the heavy oscillations in it's $S$ matrix trajectory (a). Adalora reduces the oscillations, however incorrectly identifies the rank and fails to converge due to the influence of the additional singular values.

## I   STRUCTURE PRESERVATION

The goal of this section is to clarify the formulation of Algorithm 1 in relation with the previous related literature. In particular, we want to show that the proposed algorithm can be seen as an efficient structure preservation formulation of a projected gradient flow (Koch & Lubich, 2007a) for training neural networks. In this section, to achieve full generality, we will denote with $Y_i$ either the pretrained matrices or the low-rank adapters.
As already mentioned in the previous section, gradient descent can be seen an forward Euler discretization of the gradient system

$$\dot{Y}_i = -\nabla_{Y_i}\mathcal{L}(Y_1, \ldots, Y_L), \quad i = 1, \ldots, L$$

The neural network $f_{Y_1,\ldots,Y_L}$ naturally induces a weighted graph, where nodes are neurons and weights are connections among them and for which the adjacency matrix can be written as:

$$\mathcal{Y} := \begin{bmatrix} 0 & Y_1 & 0 & \cdots & 0 \\ 0 & 0 & Y_2 & \cdots & 0 \\ \vdots & \vdots & \vdots & \ddots & \vdots \\ 0 & 0 & 0 & \cdots & Y_L \\ 0 & 0 & 0 & \cdots & 0 \end{bmatrix} \in \mathbb{R}^{|\mathcal{N}| \times |\mathcal{N}|}$$

where $|\mathcal{N}| = \sum_{i=1}^{L} d_i$ is the total number of neurons of the neural network.
The matrix $\mathcal{Y}$ now represents the adjacency matrix of the computational graph, and the block structure is given by the layers. A model with a general full adjacency matrix $\mathcal{Y}$, would in general have non-zero connections between two generic layers $i, j$, descriebed by the block $Y_{ij}$. Let's consider the model

$$f_{\mathcal{Y}}(x) = z^L(x), \quad z^0(x) = x, \quad z^{\ell+1} = \sigma_i\left(\sum_i Y_{i,\ell+1}z^i\right)$$

Notice that for $\mathcal{Y}$ upper diagonal, the previous model would be a feedforward network. Given this observation, under the assumption that $f_{\mathcal{Y}}(x)$ is well defined as an eventual fixed point, we can now

see the loss function as a function of the full adjacency matrix, with an abuse of notation we will call it again $\mathcal{L}(\mathcal{Y})$. Usual training would superimpose the sparse graph with the same structure of $\mathcal{Y}$, but let's consider for a moment the gradient flow

$$\dot{\mathcal{Y}} = -\nabla\mathcal{L}(\mathcal{Y})$$

Clearly, the flow does not preserve the sparsity of the adjacency matrix $\mathcal{Y}$, even for sparse initial conditions. Using the theory developed in (Koch & Lubich, 2007a) directly on this gradient flow would lead to neural networks with a non-feedforward topology. Moreover, given the size of $|\mathcal{N}|$ for modern neural networks, it can be expensive to compute QR or SVD decomposition of the basis matrices. Luckily, the sparsity structure is a simple linear constraint represented by the mask matrix

$$\mathcal{M} := \begin{bmatrix} 0 & 11^\top & 0 & \cdots & 0 \\ \hline 0 & 0 & 11^\top & \cdots & 0 \\ \hline \vdots & \vdots & \vdots & \ddots & \vdots \\ \hline 0 & 0 & 0 & \cdots & 11^\top \\ \hline 0 & 0 & 0 & \cdots & 0 \end{bmatrix} \in \mathbb{R}^{|\mathcal{N}|\times|\mathcal{N}|}$$

and the linear operator $\Pi(A) = \mathcal{M} \odot A$.

A system preserving the sparsity pattern is given naturally by the ODE

$$\dot{\mathcal{Y}} = -\Pi\nabla\mathcal{L}(\mathcal{Y})$$

However, it is not obvious that by projecting this last system on the manifold of rank-r matrices $\mathcal{M}_r$ the block structure is preserved. Fortunately, it is indeed the case, described by the following lemma:

**Proposition 2.** *(Block structure preservation of the flow)*
*Consider the gradient flow with sparse initial condition*

$$\dot{\mathcal{Y}} = -P(\mathcal{Y})\Pi\nabla\mathcal{L}(\mathcal{Y}), \ \ \mathcal{Y}(0) = \mathcal{Y}_0 \in range(\Pi)$$

*Then $\mathcal{Y}(t) \in range(\Pi)$ for all $t \geq 0$.*

*Proof.* It is necessary and sufficient to prove that $P(\mathcal{Y}(t))\Pi\nabla\mathcal{L}(\mathcal{Y}(t)) \in range(\Pi)$ for all $t \geq 0$, i.e. that $\Pi P(\mathcal{Y}(t))\Pi\nabla\mathcal{L}(\mathcal{Y}(t)) = P(\mathcal{Y}(t))\Pi\nabla\mathcal{L}(\mathcal{Y}(t))$. The key to prove this is to observe that for $Z \in range(\Pi)$, we have $P(\mathcal{Y})Z \in range(\Pi)$. In fact, given $Z \in range(\Pi)$, we can write a SVD of $Z$ as

$$Z = \begin{bmatrix} 0 & U_1 & 0 & \cdots & 0 \\ \hline 0 & 0 & U_2 & \cdots & 0 \\ \hline \vdots & \vdots & \vdots & \ddots & \vdots \\ \hline 0 & 0 & 0 & 0 & U_L \\ \hline I & 0 & 0 & \cdots & 0 \end{bmatrix} \begin{bmatrix} 0 & 0 & 0 & \cdots & 0 \\ \hline 0 & S_1 & 0 & \cdots & 0 \\ \hline \vdots & \vdots & \vdots & \ddots & \vdots \\ \hline 0 & 0 & 0 & S_{L-1} & 0 \\ \hline 0 & 0 & 0 & \cdots & S_L \end{bmatrix} \begin{bmatrix} I & 0 & 0 & \cdots & 0 \\ \hline 0 & V_1^\top & 0 & \cdots & 0 \\ \hline \vdots & \vdots & \vdots & \ddots & \vdots \\ \hline 0 & 0 & 0 & V_{L-1}^\top & 0 \\ \hline 0 & 0 & 0 & \cdots & V_L^\top \end{bmatrix}$$

$\square$

and we have $UU^\top, VV^\top \in range(\Pi)$. Thus, by direct calculation we can show that $UU^\top Z, ZVV^\top, UU^\top ZVV^\top \in range(\Pi)$ and thus $P(\mathcal{Y})Z = UU^\top Z + ZVV^\top - UU^\top ZVV^\top \in range(\Pi)$. Since $\Pi\nabla\mathcal{L}(\mathcal{Y}(t)) \in range(\Pi)$ by construction for all $t \geq 0$, we get the desider result. Thanks to this last proposition, following again the line of work in (Koch & Lubich, 2007a), it is possible to restrict the parameterization in the tangent space to a block-structured one as in Proposition 1. In this way, we get the following coherence theorem:

**Proposition 3.** *Consider the gradient flow with sparse initial condition*

$$U = \dot{\mathcal{Y}} = -P(\mathcal{Y})\Pi\nabla\mathcal{L}(\mathcal{Y}), \ \ \mathcal{Y}(0) = \mathcal{Y}_0 \in range(\Pi)$$

*Consider now the parametrization $\mathcal{Y} = USV^\top$ with*

$$U = \begin{bmatrix} 0 & U_1 & 0 & \cdots & 0 \\ \hline 0 & 0 & U_2 & \cdots & 0 \\ \hline \vdots & \vdots & \vdots & \ddots & \vdots \\ \hline 0 & 0 & 0 & 0 & U_L \\ \hline I & 0 & 0 & \cdots & 0 \end{bmatrix}, S = \begin{bmatrix} 0 & 0 & 0 & \cdots & 0 \\ \hline 0 & S_1 & 0 & \cdots & 0 \\ \hline \vdots & \vdots & \vdots & \ddots & \vdots \\ \hline 0 & 0 & 0 & S_{L-1} & 0 \\ \hline 0 & 0 & 0 & \cdots & S_L \end{bmatrix}, V = \begin{bmatrix} I & 0 & 0 & \cdots & 0 \\ \hline 0 & V_1 & 0 & \cdots & 0 \\ \hline \vdots & \vdots & \vdots & \ddots & \vdots \\ \hline 0 & 0 & 0 & V_{L-1} & 0 \\ \hline 0 & 0 & 0 & \cdots & V_L \end{bmatrix}$$

*where $U_i^\top U_i = I, V_i^\top V_i = I$. Then, by imposing the Gauge conditions $\dot{U}_i^\top U_i = 0, \dot{V}_i^\top V_i = 0$, the projected flow $\dot{\mathcal{Y}} = -P(\mathcal{Y})\Pi\nabla\mathcal{L}(\mathcal{Y})$ can be rewritten in block fashion as follows:*

$$\dot{S}_i(t) = -U_i^\top(t)\nabla_{Y_i}\mathcal{L}(U(t)S(t)V(t)^\top)V_i(t),$$

$$\dot{U}_i(t) = -\left(I - P_{U_i(t)}\right)\nabla_{Y_i}\mathcal{L}(U(t)S(t)V(t)^\top)V_i(t)S_i(t)^{-1},$$

$$\dot{V}_i(t) = -\left(I - P_{V_i(t)}\right)\nabla_{Y_i}\mathcal{L}(U(t)S(t)V(t)^\top)U_i(t)S_i(t)^{-\top}, \quad i = 1, \ldots, L$$

*Proof.* Thanks to the previous proposition, we know that the variation $P(\mathcal{Y})\Pi\nabla\mathcal{L}(\mathcal{Y}) \in range(\Pi)$ for all $t \geq 0$. Then, we have $\mathcal{Y}(t) \in range(\Pi)$ for all times, and thus we can decompose it using a block SVD as described in the statement of the proposition. Moreover, by the self-adjointness of $\Pi$, Galerkin condition can be written as:

$$\langle \dot{\mathcal{Y}} + \nabla\mathcal{L}(\mathcal{Y}), q\rangle = \langle \dot{U}SV^\top + U\dot{S}V^\top + US\dot{V}^\top + \nabla\mathcal{L}(\mathcal{Y}), q\rangle = 0, \quad \forall q \in T_{\mathcal{Y}}\mathcal{M}_r \cap range(\Pi)$$

Since $q \in T_{\mathcal{Y}}\mathcal{M}_r \cap range(\Pi)$, we can represent it as $q = \delta USV^\top + U\delta SV^\top + US\delta V^\top$, with $\delta U, \delta V, \delta S$ with the same block structure of $U, S$ and $V$. By writing the last conditions on a basis of $T_{\mathcal{Y}}\mathcal{M}_r \cap range(\Pi)$, we get

$$\langle \dot{U}SV^\top + U\dot{S}V^\top + US\dot{V}^\top + \nabla\mathcal{L}(\mathcal{Y}), \delta USV^\top\rangle = 0$$

$$\langle \dot{U}SV^\top + U\dot{S}V^\top + US\dot{V}^\top + \nabla\mathcal{L}(\mathcal{Y}), U\delta SV^\top\rangle = 0$$

$$\langle \dot{U}SV^\top + U\dot{S}V^\top + US\dot{V}^\top + \nabla\mathcal{L}(\mathcal{Y}), US\delta V^\top\rangle = 0$$

Thanks to the Gauge conditions $\dot{U}^\top U = 0, \dot{V}^\top V = 0$ and to the properties of the Frobenius inner product, the last system becomes

$$\langle \dot{U}SS^\top + \nabla\mathcal{L}(\mathcal{Y})VS^\top, \delta U\rangle = 0$$

$$\langle U^\top U\dot{S}V^\top V + U^\top\nabla\mathcal{L}(\mathcal{Y})V, \delta S\rangle = \langle \dot{S} + U^\top\nabla\mathcal{L}(\mathcal{Y})V, \delta S\rangle = 0$$

$$\langle S^\top S\dot{V}^\top + S^\top U^\top\nabla\mathcal{L}(\mathcal{Y}), \delta V^\top\rangle = 0$$

and from this equations we get the known

$$\dot{U} = -(I - UU^\top)\nabla\mathcal{L}(\mathcal{Y})VS^{-1}$$

$$\dot{S} = -U^\top\nabla\mathcal{L}(\mathcal{Y})V$$

$$\dot{V} = -(I - VV^\top)\nabla\mathcal{L}(\mathcal{Y})^\top US^{-\top}$$

By writing this equations block-by-block, we get the desidered result. $\square$

This last proposition clarifies how to connect the single matrix setting with the multi-matrix setting, showing that the presentation of Algorithm 1 is in fact coherent with the single matrix setting. Moreover, investigation of this setting leads naturally to the global truncation strategy.

## I.1 TRUNCATION STRATEGY

The global truncation strategy proposed in the main manuscript is in fact coherent with the single matrix formulation presented in the previous section. In fact, one can assemble the rank augmented $S$ matrix as:

$$\widehat{S}(t = 1) = \begin{bmatrix} \widehat{S}_1 & 0 & 0 & \cdots & 0 \\ 0 & \widehat{S}_2 & 0 & \cdots & 0 \\ \vdots & \vdots & \vdots & \ddots & \vdots \\ 0 & 0 & 0 & \widehat{S}_{L-1} & 0 \\ 0 & 0 & 0 & \cdots & \widehat{S}_L \end{bmatrix}$$

and then truncate the smallest singular values up to required precision. This can be efficiently done by computing an SVD on each diagonal block, giving effectively an SVD of the global matrix. In particular, if $\widehat{S}_i = P_i\Sigma_i Q_i^\top$ we get that

$$\widehat{S} = blockdiag(P_1, \ldots, P_L)blockdiag(\Sigma_1, \ldots, \Sigma_L)blockdiag(Q_1, \ldots, Q_L)^\top$$

Since the matrix $blockdiag(\Sigma_1, \ldots, \Sigma_L)$ is effectively diagonal, by assuming the diagonal is increasingly ordered, it is natural to globally truncate the ranks according to the minimal $k$ such that

$$\frac{\sum_{i=k+1}^{2rL} \sigma_i^2}{\sum_{i=1}^{2rL} \sigma_i^2} < \frac{\tau}{1 - \tau}$$

Which corresponds in throwing away the smallest singular values of $\widehat{S}$ until we reach the desired relative error. By rewriting this criterion on the singular values of each matrix $\widehat{S}_i$, we get exactly the global criterion proposed in Section 4.

## J    OPTIMALITY ON THE LOW-RANK MANIFOLD

We remark below that if $W_\star$ is a local minimum, then $P(W_\star)\nabla\mathcal{L}(W_\star) = 0$. In particular, since $P(W)\nabla\mathcal{L}(W)$ is the Riemannian gradient with respect to the ambient metric, then the following holds by definition of the gradient:

$$\partial_{\delta W}\mathcal{L}(W) = \langle P(W)\nabla\mathcal{L}(W), \delta W \rangle$$

where $\partial_{\delta W}\mathcal{L}(W)$ is the directional derivative of $\mathcal{L}$ along the direction $\delta W$. Thus, $P(W_\star)\nabla\mathcal{L}(W_\star) = 0$ if and only if $\partial_{\delta W}\mathcal{L}(W) = 0$ for all $\delta W \in T_W\mathcal{M}$ and this happens if and only if $\nabla\mathcal{L}(W) \in (T_W\mathcal{M})^\perp$. So geometrically, if $W_\star$ is a local minimum, then $P(W_\star)\nabla\mathcal{L}(W_\star) = 0$ means that among all available directions, there are none that decrease the loss.

For simultaneous descent, the same condition doesn't hold, in fact, the algorithm's stationary points satisfy $\widehat{P}(W_\star)\nabla\mathcal{L}(W_\star) = 0$, which given the non-orthogonality does not, in general, imply $P(W_\star)\nabla\mathcal{L}(W_\star) = 0$, so there could be descent directions unexploited by the method.

