# OpenReview forum: "GeoLoRA: Geometric integration for parameter efficient fine-tuning"
_ICLR.cc/2025/Conference — ICLR 2025 Poster_

### Official Review · Reviewer_Ht32 · 2024-11-02

**Soundness:** 2
**Presentation:** 3
**Contribution:** 2
**Rating:** 5
**Confidence:** 2

**Summary:**

This paper proposes a new LoRA variant, GeoLoRA, which can adaptively allocate rank budget and reduce the computational overhead. They analyze the limitation of low-rank optimization  and imporve LoRA according to the Riemannian gradient flow analysis. They also provide theoretical results about convergence and error bounds. The experiments show that GeoLoRA can improve accuracy and training speed compared to AdaLoRA.

**Strengths:**

1. The analysis of Section 3 clearly point out the limitation of current low-rank optimization, i.e., the optimality condition for low-rank optimization is that the projection of the gradient to 0. Further analysis shows that parallel optimization may not be able to achieve such optimality condition, indicating the urgency of more advanced methods.
2. The paper describes their methodology very clearly. The readers can easily understand how GeoLoRA is implemented.

**Weaknesses:**

1. GeoLoRA needs to be validated on larger datasets and models, such as MetaMathQA task and  LLAMA. The authors claim GeoLoRA can converge to an optimal low-rank solution, so they should evaluate GeoLoRA under the settings where LoRA's performance is worse than full finetuning to validate their claim.
2. The improvements seems to be minor, in some subsets of GLUE, GeoLoRA performs even worse than vanilla LoRA. Moreover, the authors should include more baseline methods (DoRA, LoRA+, PiSSA) in their experiments to show the necessity of adaptive rank allocation (TABLE 2,3,4).
3. The paper lacks an intuitive explanation of their method. Although there are some formal connections between  Section 3 and Section 4, it's not clear to me why the authors design GeoLoRA. Thus, the theoretical contribution of GeoLoRA is not clear.

I would consider increasing the score if the authors can explain the methods more clearly and provide more solid experiments.

**Questions:**

1. How to choose a proper $\tau$ and initial rank for GeoLoRA? In Table 7, it seems that you chose different $\tau$ and initial rank for different dataset. Does this suggest that GeoLoRA is sensitive to hyperparameters on visual experiments?
2. The authors provide a toy example to show the  behaviours of different low-rank adaptation strategies. How does they behave in more practical settings? (for example, the settings of Table 2)

---

> ### Author Response · Authors · 2024-11-16
> **Answer to reviewer Ht32**
>
> We thank the reviewer for their constructive feedback. We are pleased that the strengths highlighted by the reviewer align with the aspects we prioritized when writing our article, and we are glad these points came through clearly. We hope our responses to the weaknesses will convince you to raise your score and accept our work.
>
> 1. **GeoLoRA validation**:
>
> We recognize the importance of larger-scale evaluations; however, given our computational resources and the fact that we average our results over multiple runs, we believe the chosen benchmarks are already considerably large.
>  We also wish to point out that a main drawback of current LoRA-type adapters is their sensitive choice of hyperparameters, which GeoLoRA does not exhibit. We have illustrated this behavior in the Ablations Section on page 9 as well as in Figure 2. Here, using hyperparameters where AdaLoRA fails to reach satisfactory accuracy, GeoLoRA can still reach good results. This is a clear example of a setting where AdaLoRA fails, whereas GeoLoRA shows good performance.
>
> 2. **Baselines for GLUE**:
>
> Please note that these benchmarks are multi-objective. For clarity, we provide the average test score across all GLUE benchmarks for each method, where it becomes apparent that GeoLoRA has the highest average score, even when the competitors have a much higher parameter count. See Table 1 in the right columns. Beyond accuracy, other important metrics include parameter count and runtime per training and evaluation step. For instance, in the QQP benchmark, GeoLoRA does not achieve the highest accuracy; AdaLoRA achieves an improved accuracy (90.65 vs. 90.53). However, AdaLoRA requires nearly twice as many parameters (1.87 times more) and is slower in both training and evaluation. Thus, despite its accuracy advantage, AdaLoRA falls short of GeoLoRA in all other metrics for the QQP benchmark. As you can see from our new numerical experiments with fine-tuned rank budgets for AdaLoRA, its accuracy for QQP is smaller than that of GeoLoRA when a similar parameter count is chosen.
>
> For the chosen baselines, we aimed to include the most widely used low-rank adapters currently available. Notably, AdaLoRA is among the most frequently used rank adaptive adapters at present. In addition, we have now added results for DoRA and LoRA+, since these approaches have openly available source code. The results further highlight the advantages of GeoLoRA, and we thank the reviewer for pushing us to make an effort to compare against these further baselines.

---

> > ### Author Response · Authors · 2024-11-16
> > **Answer to reviewer Ht32, part 2**
> >
> > Please find the table with the extended GLUE benchmarking results below:
> >
> > We report target metrics and computational performance (higher is better) for the median of 5 runs
> > using different random seeds. Best results per dataset are shown in bold.
> >
> > | **Method** (_# Params_)          | **MNLI** (Acc)    | **SST-2** (Acc)     | **CoLA** (Mcc)      | **QQP** (F1)      | **QNLI** (Acc)  | **RTE** (Acc)   | **MRPC** (Acc)      | **STS-B** (Corr)    | **Mean**            |
> > |----------------------------------|-------------------|---------------------|---------------------|-------------------|-----------------|-----------------|---------------------|---------------------|---------------------|
> > | **Full FT** (_184M_)             | 89.90             | 95.63               | 69.19               | 89.80             | 94.03           | 83.75           | 89.46               | 91.60               | 87.92               |
> > | **BitFit** (_0.1M_)              | 89.37             | 94.84               | 66.96               | 84.95             | 92.24           | 78.70           | 87.75               | 91.35               | 85.77               |
> > | **HAdapter** (_1.22M_)           | 90.13             | 95.53               | 68.64               | 89.27             | 94.11           | 84.48           | 89.95               | 91.48               | 87.94               |
> > | **PAdapter** (_1.18M_)           | 90.33             | 95.61               | 68.77               | 89.40             | **94.29**       | 85.20           | 89.46               | 91.54               | 88.07               |
> > | **LoRA r=8** (_1.33M_)           | 90.29             | 95.29               | 68.57               | 90.61             | 93.91           | 85.50           | 89.75               | 89.10               | 87.87               |
> > | **LoRA+ r=8** (_1.33M_)          | *90.39*           | *95.37*             | *70.56*             | *90.65*           | *94.05*         | *85.59*         | *89.65*             | *88.07*             | *88.04*             |
> > | **DoRA r=8** (_1.33M_)           | *90.11*           | *94.30*             | *68.50*             | **90.71**         | *94.31*         | *85.05*         | *89.32*             | *91.38*             | *87.96*             |
> > | **AdaLoRA target r=8** (_1.27M_) | **90.44**         | 95.64               | 68.76               | *90.65*           | 94.11           | **86.00**       | 89.44               | 91.41               | 88.30               |
> > | **AdaLoRA (matched parameters)** | *90.21* (_0.75M_) | 95.64 (_1.27M_)     | 68.59 (_1.07M_)     | *90.48* (_0.72M_) | 93.93 (_0.72M_) | 85.92 (_1.16M_) | 88.21 (_0.74M_)     | 90.91 (_0.74M_)     | *88.28* (_0.89M_)   |
> > | **GeoLoRA**                      | *90.38* (_0.7M_)  | **95.98** (_1.17M_) | **69.03** (_0.98M_) | *90.53* (_0.69M_) | 94.23 (_0.70M_) | 85.93 (_1.19M_) | **90.10** (_0.75M_) | **91.58** (_0.71M_) | **88.47** (_0.86M_) |

---

> > > ### Author Response · Authors · 2024-11-16
> > > **Answer to reviewer Ht32, part 3**
> > >
> > > 3. **Intuition of the Method**:
> > >
> > > Frankly, we are surprised by the conclusion that the presentation of the method and our theoretical contribution are unclear, especially since our paper provides a clear motivation (section 3) and a simple presentation of the method, as well as a structured and rigorous derivation of its theoretical properties.
> > >
> > > As is often the case with advanced computational algorithms, a deeper technical intuition behind the method is strictly linked with the proofs of its theoretical properties (Theorems 2 and 3), which we have detailed in the supplementary materials due to space constraints. We agree that deriving a method through its theoretical analysis may somewhat lack immediate intuitive simplicity, but it is a valid and powerful approach to ensure robust theoretical underpinnings and is certainly not uncommon. We are thus surprised that this approach has led to concerns regarding the clarity or relevance of the theoretical contribution and would appreciate specific feedback on which aspects of our theoretical contribution remain unclear.
> > >
> > > 4. **Ablation Study**:
> > >
> > > We address this question in the "Ablations" section of our manuscript (see page 9), where we conduct a detailed investigation of how hyperparameters influence overall performance. Note that in Figure 2 we have studied and presented the method's behavior and comparison to AdaLoRA, which is a direct answer to your question. While there is some dependence on hyperparameters, GeoLoRA shows robust, satisfactory results over a considerable interval of rank budget and learning rate.  This is a stark contrast to AdaLoRA and one of the advantages of our method.  Further, in Figure 2 b) we specifically investigate the effect of the initial rank for GeoLoRa and find that the method is fairly robust with respect to this choice: The rank augmentation and truncation is able to recover the true rank of the network regardless of the initalization rank.
> > >
> > > 5. **Behavior of low-rank adaptation strategies**:
> > >
> > > The test case in Section~3 was designed to illustrate potential issues with standard training through a straightforward numerical example that most readers are familiar with. Due to the stochastic effects in more complex settings that require reduced batch sizes, we expect a much harder-to-interpret behavior, which is why we chose a simple, deterministic setting. However, as shown in our numerical examples in Section 5, we observe a reduced efficiency of simultaneous descent methods in these settings as well.

---

> ### Comment · Reviewer_Ht32 · 2024-11-16
> **Comments on the Authors' Rebuttal**
>
> Thanks for your response. I still have the following concerns:
>
> ### About Experiments
>
> 1. **Larger-scale evaluations**:  I still believe that evaluating LoRA variants on larger datasets and models is crucial, since LoRA was originally proposed to address the substantial resource demands involved in full fine-tuning of LLMs. Based on my experience, training LLAMA-7B on a 10k subset of MetaMathQA takes approximately 7 hours using a single NVIDIA 4090. Given that there is still over a week left before the rebuttal period ends, I encourage the authors to provide additional experiments.
>
>
> 2. **Hyperparameter Selection**: Figure 2(a) and 2(b) demonstrate that GeoLoRA can converge to the intrinsic rank with different initial ranks, doesn't this indicate that GeoLoRA is insensitive to the initial rank? The authors did not adjust the initial rank or $\tau$ in Table 6, so why did they adjust the initial rank in Table 7? Additionally, the question of $\tau$ selection remains unaddressed.
>
>     This issue is not about ablation study; it's about the practicality of GeoLoRA (as a meticulous hyperparameter search would increase computational costs) and the fairness of comparisons. Moreover, when comparing Table 7 with Table 6, I noticed that Table 6 did not modify the initial rank or $\tau$, but it used a different learning rate, which is not mentioned in section B.1.2, along with a different number of epochs (potentially due to early stopping based on the validation set?).
>
>     I hope the authors could clarify the questions about (1) why they chose different initial ranks even GeoLoRA is not sensitive to the initial rank (2) how GeoLoRA behaves with different $\tau$ (3) why they chose a learning rate not mentioned for other methods, does this imply the results of GeoLoRA in Table 2 are obtained with an unfair hyperparameter selection?
>
>
> ### About Theory
>
> Apologies for not expressing my confusion clearly in the previous round. I must admit that I am not very familiar with the theoretical aspects of the paper. The low confidence is mainly due this (the low score is mainly due to the experiments).
>
> I agree that Section 3 provides a clear motivation and there is a solid link between Section 3 and Eq. (6).  Personally, I did not find the theoretical motivations for the augmentation and truncation operations. Are these operations designed intuitively for dynamic rank allocation, or is there a deeper theoretical motivation behind them?
>
>
> ### Minor Question
>
> **Behavior of low-rank adaptation strategies**: I agree that the behavior of these strategies can be more complex in practical experiments. If the authors have the original experimental records, providing the training loss curves would be an effective way to demonstrate that GeoLoRA converges more efficiently than other methods.

---

> > ### Author Response · Authors · 2024-11-22
> >
> > Thank you for your feedback. We are pleased to see that you no longer have concerns about the quality of the improvements or comparisons to additional baselines. However, we understand that you remain concerned about (1) the size of the benchmarks and (2) the choice of hyperparameters, which you are afraid could lead to a potentially unfair comparison with baseline methods.
> >
> > Let us resolve your concerns regarding hyperparameters directly. We acknowledge that our original manuscript could have provided better explanations, as some hyperparameters were initially selected without any search method, just following intuition. In particular, in Table 7, we expect Cifar100 and Tiny-Imagenet to be harder tasks compared to Cifar10, and thus the higher starting rank for the adapters. However, as you correctly mentioned, this intuition is not needed since GeoLoRA shows little dependence on initial ranks.
> > To address your concerns, we have now redone all numerical experiments where this matter was present, by setting the same initial rank. These updated results reinforce our findings from the ablation study and demonstrate that GeoLoRA exhibits a significantly reduced dependence on the initial rank.
> >
> > We would also like to remark that we did not perform any hyperparameter search to cherry-pick optimal hyperparameters for our method. Instead, we deliberately chose the same starting rank and learning rate from baseline methods to ensure a fair comparison. Moreover, the batch size and number of training epochs is consistent across all methods in each experiment. To be more specific and directly answer to your questions:
> >
> > 1. **Why they chose different initial ranks?**: The choice of initial rank for the Vision Transformer benchmark is somewhat heuristic and discussed above and reported for completeness. As you correctly stated, GeoLoRA is relatively insensitive to this choice, as reported in Figure 2. To resolve your concerns and improve the presentation, we have repeated the Vision Transformer runs of Table 3 for ranks 10, 16, and 32 for all datasets (Cifar10, Cifar100, and Tiny Imagenet). The resulting validation accuracy values are within $0.03\%$ deviation of the reported values in the table. The resulting parameter count deviation is below $2\%$ of the reported values, effectively not changing the results in the table. This finding is in line with the observations of Figure 2.
> >
> > | Method                             | Cifar 10 [\%] | Cifar 100 [\%] | Tiny-Imagenet  [\%] |
> > |------------------------------------|------------------------------------|-------------------------------------|------------------------------------------|
> > |                                    | \# Params                          | Acc [\%]                            | \# Params                                | Acc [\%]                 | \# Params                     | Acc [\%]                |
> > | LoRA             | 0.47M (r=3)      | 98.47             | 0.47M (r=3)           | 91.47 | 0.99M (r=6) | 87.34 |
> > | AdaLoRA                            | 0.47M                     | 98.51                               | 0.45M                                    | 91.44                    | 0.9M                          | 87.21                   |
> > | GeoLoRA, local                     | 0.47M                     | 98.55                      | 0.35M                           | 91.63           | 0.92M                         | 88.09          |
> > | GeoLoRA, global                    | 0.48M                              | 98.51                               | 0.47M                                    | 91.62                    | 0.75M                | 88.07                   |
> > | GeoLoRA, local (r=10, $\tau=0.15$) | 0.472M                           | 98.52                             | 0.351M                                 | 91.60                  | 0.904M                        | 88.08                 |
> > | GeoLoRA, local (r=16, $\tau=0.15$) | 0.472M                           | 98.55                             | 0.357M                                 | 91.63                  | 0.909M                        | 88.10                 |
> > | GeoLoRA, local (r=32, $\tau=0.15$) | 0.473M                           | 98.54                             | 0.362M                                 | 91.63                  | 0.921M                        | 88.09                 |

---

> > > ### Author Response · Authors · 2024-11-22
> > >
> > > 2. **How does GeoLoRA behave with different $\tau$?**: The compression tolerance $\tau$ can be chosen to define the desired compression rate of the network, similar to the rank that needs to be chosen for fixed-rank methods or the rank budget chosen in AdaLoRA. It, therefore, has an impact on the results (though the method is not highly sensitive to $\tau$ and a $\tau$ in the range of $0.1$ often gives good compressions with sufficient accuracy). Of course, the dependence on $\tau$ is a desired property and not surprising. We remark that unlike in LoRA, which requires a choice of the rank in every layer, only one $\tau$ needs to be chosen for the entire network. The parameter has a directly interpretable meaning as it identifies the threshold at which singular values are set to zero. Also, combining GeoLoRA with AdaLoRA's rank budget strategy is straightforward. Hence, our method is not limited to the proposed truncation strategy that uses $\tau$. Note that we present the result for the vision transformer with both truncation strategies, namely by using $\tau$ (called global) and by using a rank budget (called local, where we choose the same rank budget as for AdaLoRA). We hope this discussion clarified the question about the selection of $\tau$, and if not, please let us know.
> > > In any case, we can include a discussion regarding this matter in the appendix.
> > >
> > > 3. **Why they chose a learning rate not mentioned for other methods, does this imply the results of GeoLoRA in Table 2 are obtained with an unfair hyperparameter selection?**: We understand the confusion regarding the learning rates, as there was a copy-paste error in this section of the appendix. On page 14 of the appendix, in the description of the AdaLoRA setup, we mistakenly included the wrong set of learning rates. These rates were not used in our implementation (neither for AdaLoRA nor for GeoLoRA).
> > > Instead, the learning rates for AdaLoRA and GeoLoRA (as well as LoRA+ and DoRA) were selected from the set: $\{8e-4, 5e-4,1e-3,1.2e-3,2.2e-3\}$, as reported in Table 6. These values were directly taken from the AdaLoRA GitHub repository: https://github.com/QingruZhang/AdaLoRA/tree/d10f5ebee16c478fa2f41a44a237b38e8c9b0338/NLU/scripts.
> > > We wish to emphasize that no hyperparameter studies were conducted to identify optimal parameters for our method. The selected learning rates may have been optimized for AdaLoRA but were not tuned for GeoLoRA. We believe this showcases also the stability of our method with respect to hyperparameter choice.
> > >
> > > In light of these answers, we have revised Section B.1.2 to improve our presentation, present the hyperparameters for our newly added results, and correct mistakes in our old hyperparameter definition. Moreover, we added a sentence that no hyperparameter study was conducted for GeoLoRA, and we used the hyperparameters as defined in the AdaLoRA code on GitHub. We hope this aspect of our paper is clear now. Please let us know if there are any further questions that we have not answered.
> > >
> > > **About Theory**
> > > From a theoretical standpoint, the augmentation of the basis (7) ensures that the augmented basis spans the full-rank weights accurately (see Step 1 on page 23 in the proof of Theorem 3). The augmentation of the coefficient matrix (8) is chosen to approximate the projected full-rank weights accurately (see Step 2 on page 23 of the proof to Theorem 3). Giving a directly understandable, intuitive explanation is complicated. One intuitive idea is that one includes the old basis in (7) since the old information will usually remain dominant after a step of the optimization method. By enlarging the basis, we enlarge the search space and enable rank-adaptivity. The construction of the augmented coefficient matrix then incorporates the information gathered in the $S$, $K$, and $L$ steps of equations (6). Here, to not include the same information twice, it is important to remove the information in $K$ and $L$ that is already encoded in the old basis $U_0$ and $V_0$ through a projection with $\tilde U$ and $\tilde V$.
> > >
> > > **Minor question**
> > > We have now reported the loss for VIT on CIFAR100 for GeoLoRA and AdaLoRA in section B4 in the appendix of the revised manuscript. As we can see in that picture, AdaLoRA might fail if the learning rate is not small enough, exactly as in the matrix completion case presented as a motivating example at the beginning of the manuscript (and as it is also showcased in Figure 2 c,d, where it is shown that AdaLoRA fails for certain learning rates). We remark that AdaLoRA performs well for well-tuned learning rates, but the range of "good" learning rates is bigger for GeoLoRA in comparison with AdaLoRA.

---

> > > > ### Comment · Reviewer_Ht32 · 2024-11-25
> > > >
> > > > Thank you for your response, which has addressed most of my concerns. However, the lack of experiments on larger-scale datasets and models still raises concerns about the scalability of GeoLoRA. Therefore, I am increasing my score to 5.

---

### Official Review · Reviewer_XLNG · 2024-11-03

**Soundness:** 4
**Presentation:** 4
**Contribution:** 3
**Rating:** 6
**Confidence:** 2

**Summary:**

This paper proposes a new parameter efficient fine-tuning (PEFT) technique based on low-rank adaption with adaptive rank allocation. Existing PEFT methods with adaptive rank allocation either lack optimality guarantees or require multiple backward passes per step---the proposed method, GeoLoRA, only requires a single gradient tape per step and comes with optimality guarantees. GeoLoRA does this by using the low-rank geometry of the adapters and work on matrix differential equations. Empirically, GeoLoRA is more efficient than existing adaptive PEFT techniques and yields higher performance on a variety of benchmarks.

**Strengths:**

- The paper is well-written and the technique is overall well-motivated.
- The theoretical support for the proposed method is solid -- the majority of the theoretical results are built on the single-layer case, but Proposition 1 extends this to the general multilayer case. If I understand this result correctly, this implies that GeoLoRA can obtain the optimal rank configuration for arbitrary architectures -- this seems like a particularly strong result to me.
- GeoLoRA appears to robustly outperform or match existing PEFT methods, and it also seems faster in terms of wall-clock time compared to existing adaptive PEFT techniques.
- GeoLoRA is robust to changes in the hyperparameters, which is already a substantial improvement over existing techniques.

**Weaknesses:**

- This paper could benefit from more discussion of the details of existing methods, and why the best adaptive methods require multiple gradient tapes and what types of guarantees they have. This information is currently summarized in a table.
- In Table 2, it appears that the same rank allocation configuration is used for every dataset, for each of the baselines, whereas the parameter count of the proposed method varies for different datasets. Can the authors comment on this?
- In Section 4.1, the authors claim that GeoLoRA can be used for low-rank pre-training, but unless I've missed something, the authors do not present any empirical validation of this claim.

**Questions:**

I am confused about the results in Table 2. Why do adaptive methods like AdaLoRA only have one parameter count for all datasets? Is this the budget, or is the same configuration used for all datasets?

---

> ### Author Response · Authors · 2024-11-16
> **Answer to reviewer XLNG**
>
> We sincerely thank the reviewer for their thoughtful and detailed evaluation of our manuscript. We have carefully addressed your questions in our revisions and hope the updates clarify these aspects and further highlight the strengths of our approach.
>
> 1. **Discussion of existing methods**:
>
> Thank you for pointing this out. The standard way to enforce geometry when optimizing over the low-rank manifold is to project the gradient flow on the tangent plane and then perform a discrete descent step followed by a retraction. This is essentially what the classical Riemannian gradient descent does. DLRT, the method we refer to in Table 1, is a rank-adaptive and stochastic version of Riemannian gradient descent that was specifically developed for training large networks with low-rank weights.  However, a well-known problem of this classical approach is that the projection and retraction steps require a serial update of the matrix factors $U$, $S$, $V$, requiring three sequential gradient computations. With GeoLoRA we overcome this major computational bottleneck while maintaining the same strong theoretical guarantees as classical approaches. If you believe this additional context would improve the quality of the presentation, we are happy to add it to the revised paper. Please let us know if you need a more detailed explanation, which we will be happy to provide.
>
> 2. **Parameters in Table 2**:
>
> You are absolutely correct. The hyperparameter controlling the number of parameters remains unchanged across different datasets, both for the baseline methods and for our proposed approach. One key advantage of our method, however, is that with a fixed hyperparameter, it automatically selects a different number of parameters depending on the dataset. As a result, you will observe varying parameter counts for GeoLoRA, even though the truncation tolerance $\tau$ is consistently set to $\tau = 0.15$. For completeness, we manually tuned the rank budget of AdaLoRA to approximately match the parameter count of GeoLoRA to provide a direct comparison of the methods. We see, that the proposed GeoLoRA achieves higher test scores than AdaLoRA for similar parameter counts. We display the extended Table below:
>
> | **Method** (_# Params_)          | **MNLI** (Acc)    | **SST-2** (Acc)     | **CoLA** (Mcc)      | **QQP** (F1)      | **QNLI** (Acc)  | **RTE** (Acc)   | **MRPC** (Acc)      | **STS-B** (Corr)    | **Mean**            |
> |----------------------------------|-------------------|---------------------|---------------------|-------------------|-----------------|-----------------|---------------------|---------------------|---------------------|
> | **Full FT** (_184M_)             | 89.90             | 95.63               | 69.19               | 89.80             | 94.03           | 83.75           | 89.46               | 91.60               | 87.92               |
> | **BitFit** (_0.1M_)              | 89.37             | 94.84               | 66.96               | 84.95             | 92.24           | 78.70           | 87.75               | 91.35               | 85.77               |
> | **HAdapter** (_1.22M_)           | 90.13             | 95.53               | 68.64               | 89.27             | 94.11           | 84.48           | 89.95               | 91.48               | 87.94               |
> | **PAdapter** (_1.18M_)           | 90.33             | 95.61               | 68.77               | 89.40             | **94.29**       | 85.20           | 89.46               | 91.54               | 88.07               |
> | **LoRA r=8** (_1.33M_)           | 90.29             | 95.29               | 68.57               | 90.61             | 93.91           | 85.50           | 89.75               | 89.10               | 87.87               |
> | **LoRA+ r=8** (_1.33M_)          | *90.39*           | *95.37*             | *70.56*             | *90.65*           | *94.05*         | *85.59*         | *89.65*             | *88.07*             | *88.04*             |
> | **DoRA r=8** (_1.33M_)           | *90.11*           | *94.30*             | *68.50*             | **90.71**         | *94.31*         | *85.05*         | *89.32*             | *91.38*             | *87.96*             |
> | **AdaLoRA target r=8** (_1.27M_) | **90.44**         | 95.64               | 68.76               | *90.65*           | 94.11           | **86.00**       | 89.44               | 91.41               | 88.30               |
> | **AdaLoRA (matched parameters)** | *90.21* (_0.75M_) | 95.64 (_1.27M_)     | 68.59 (_1.07M_)     | *90.48* (_0.72M_) | 93.93 (_0.72M_) | 85.92 (_1.16M_) | 88.21 (_0.74M_)     | 90.91 (_0.74M_)     | *88.28* (_0.89M_)   |
> | **GeoLoRA**                      | *90.38* (_0.7M_)  | **95.98** (_1.17M_) | **69.03** (_0.98M_) | *90.53* (_0.69M_) | 94.23 (_0.70M_) | 85.93 (_1.19M_) | **90.10** (_0.75M_) | **91.58** (_0.71M_) | **88.47** (_0.86M_) |

---

> > ### Author Response · Authors · 2024-11-16
> > **Answer to reviewer XLNG, part 2**
> >
> > 3. **Low-Rank Pretraining**:
> >
> > This statement applies not only to GeoLoRA but also to most (if not all) methods for low-rank adapters. Any training method designed for low-rank factorization can be directly adapted to low-rank pre-training by initializing the pre-trained weights to zero. We included this point because we observed a divergence in the literature between low-rank pre-training and low-rank adapters, despite the potential for these fields to benefit directly from each other. If you find this statement unclear, we would be open to removing it from our manuscript.

---

### Official Review · Reviewer_MVJ1 · 2024-11-04

**Soundness:** 3
**Presentation:** 3
**Contribution:** 3
**Rating:** 8
**Confidence:** 3

**Summary:**

The popular finetuning method LoRA is learned as an optimization problem with some variant of gradient descent. However, these may not arrive at the optimal low-rank solution. This issue has been looked into before, but that solution provides a stiff gradient flow. This work first reparameterizes the gradient flow to be non-stiff. Further, GeoLoRA allows for the rank of the LoRA adapters to vary, eliminating the need to specify that hyperparameter. This parameterization only requires a single backward pass rather than multiple which are needed by other methods. Based on a clear theoretical backing, theoretical results follow, with loss descent being guaranteed.

**Strengths:**

- This work provides a novel way of solving the stiffness/convergence issues in existing LoRA solvers, finding the LoRA factors at around the same rate as full fine-tuning.
- GeoLoRA, while being more efficient than standard LoRA training, also includes adaptive rank finding, which is the most common alternative AdaLoRA has been shown to do but at a significantly greater cost than GeoLoRA. Finding LoRA-like fine-tuning factors can be done without specifying the rank a prior at nearly the same if not fewer training iterations.
- Experiments showing GeoLoRA converging to around the same number of trainable parameters, with both starting with lesser and greater parameters, depict how this method does work towards decreasing rank if the extra dimensions are unneeded.
- Timing experiments showing a direct efficiency improvement over AdaLoRA in Table 2 strongly defend that this method is an improvement over its alternatives.

**Weaknesses:**

- For clarity, the motivation (Section 3) is currently a little lengthy. When reading, there was some expectation that one of these major derivations was part of the GeoLoRA algorithm, and it only became clear that the algorithm's description happened a few pages later. Including either a more complete description of Section 3 at its beginning, another subsection title, or moving some of Section 3 to the appendix/another section would help with readability.

**Questions:**

- Determining if the rank should be changed is dependent on calculating orthonormalizations of n-by-2r0-sized matrices and SVDs of r-by-r matrices. With r significantly less than n, is it possible to use approximate methods such as power iteration to accelerate GeoLoRA further?

---

> ### Author Response · Authors · 2024-11-16
> **Answer to reviewer MVJ1**
>
> Thank you for taking the time to carefully review our work and for your positive feedback. We have addressed the questions and comments you raised.
>
> 1. **Clarity of Section 3**:
>
> We agree that we have dedicated a lot of space to Section 3. We felt this discussion was essential, as many researchers are unaware of this behavior, which is currently missing from the literature on low-rank adapters. To improve the structure of our article, we have added a sentence explaining the intention of Section 3 in our introduction, where we now highlight the structure of our article and point out that the main method is proposed in Section 4.
>
> 2. **Question about rank determination**:
>
>  Thank you for this interesting question. We believe this is an interesting idea that could be investigated in future research. Further research on this question could also explore the use randomized SVDs and QR decompositions.

---

### Official Review · Reviewer_kJ1Q · 2024-11-04

**Soundness:** 2
**Presentation:** 3
**Contribution:** 3
**Rating:** 6
**Confidence:** 3

**Summary:**

This paper come up with a more efficient and robust variant of low-rank adaptation, GeoLoRA. The intuition is straightforward — existing implementation of low-rank adaptation using gradient descent, which update U, S, V together and the update of S cause the algorithm slow to converge.

This method proposed GeoLoRA, which integrates rank adaptivity, low-rank optimality, and memory and computational efficiency. This method controls the update of S, supported by theoretical convergence guarantee and error analysis, and various experiment results.

**Strengths:**

- [Major] Easy to follow, well-written
- [Major] The intuition is straightforward, and the method is simple yet effective
- [Major] The proposed method is supported by many evidence, including theoretical convergence guarantee and error analysis, and various experiment results.
- [Major] The experiments are diverse, including both language tasks and vision tasks.

**Weaknesses:**

- [Major] While the figure between L222 and L232 exhibit great performance of GeoLoRA, the improvement caused by GeoLoRA seems to be marginal, especially in Table 2.
- [Major] Some experiment settings are not well-justified. For instance, in Table 2, the authors report the experiment results of GeoLoRA individually, while for others, it seems not. It looks like an unfair comparison, and the reason why GeoLoRA are reported individually should be justified. For Table 3, the results of LoRA are missing; for Table 4, the results of AdaLoRA are missing. More discussion are needed to justify the experiment results.
- [Medium] More baselines are needed, since there exists many LoRA variants.

Personally speaking, I like this paper. However, given the current limitations of the experiments, I can only give 5. If authors are able to address my concerns listed above, I am willing to increase my score to accept this paper.

**Questions:**

See the weakness section.

**Details Of Ethics Concerns:**

No.

---

> ### Author Response · Authors · 2024-11-15
> **Answer to reviewer kJ1Q**
>
> Thank you for carefully reviewing our work and for your feedback. We have addressed the questions and comments you raised, and we hope the revisions, clarifications, and the new results we have included will address your concerns and convince you to accept our work.
>
> 1. **Performance of GeoLoRA**:
>
> The test case in Section~3 was designed to highlight potential issues with standard training through a simple numerical
> example that is easily understandable for readers. While we agree that the results are less pronounced for more complex
> benchmarks, we would like to emphasize that GLUE is a multi-objective benchmark, and we believe GeoLoRA performs
> exceptionally well. For clarity, we provide the average test score across all GLUE benchmarks for each method, where it
> becomes apparent that GeoLoRA has the highest average score, even when the competitors have a much higher parameter
> count. See Table 1 in the right columns.
>
> In addition to accuracy, we emphasize that GeoLoRA shows remarkably superior performance against other key metrics,
> including the number of parameters and the runtime per training step and evaluation. For instance, in the QQP benchmark,
> GeoLoRA does not achieve the highest accuracy; AdaLoRA achieves an improved accuracy (90.65 vs. 90.53). However, AdaLoRA
> requires nearly twice as many parameters (1.87 times more) and is slower in both training and evaluation.
>
> Thus, despite its accuracy advantage, AdaLoRA falls short of GeoLoRA in all other metrics for the QQP benchmark. We have
> added AdaLoRA runs, where we fine-tuned the rank budget so that AdaLoRA exhibits a parameter count similar to GeoLoRA.
> This again underlines the improved performance of our method. For example, AdaLoRA's accuracy for QQP becomes smaller
> than GeoLoRA's when a similar parameter count is chosen. In addition, we would like to remark that it is often observed
> that standard LoRA-type adapters are very sensitive to hyperparameters, which is not the case for GeoLoRA, see our
> ablations study on page 9 as well as Figure 2.
>
> 2. **Experiment Settings**:
>
> Thank you for requesting clarification. Since reviewer XLNG raised the same question, we believe adding an explanation
> of why GeoLoRA has a different number of parameters for each architecture would strengthen our manuscript. This aspect
> is important, as it highlights a key advantage of GeoLoRA. We train all benchmarks using the same compression
> hyperparameter, meaning that it is not fine-tuned for each individual GLUE benchmark. While most architectures use a
> fixed rank, resulting in a constant number of parameters across benchmarks, AdaLoRA is rank-adaptive. However, the way
> the rank budget is allocated in AdaLoRA also leads to a constant parameter count for each benchmark. In contrast, a main
> distinction of GeoLoRA is a higher degree of adaptation and flexibility, highlighted by its ability to automatically
> select a different number of parameters for each benchmark, all while using the same compression hyperparameter.
>
> For completeness, we manually tuned the rank budget of AdaLoRA to approximately match the parameter count of GeoLoRA to
> provide a direct comparison of the methods (see AdaLoRA, matched parameters in Table 1). We see that the proposed
> GeoLoRA achieves higher test scores than AdaLoRA for similar parameter counts.
>
> Further, we have added LoRA in the Vision transformer benchmarks; see Table 2. We set the rank $r=3$ for Cifar10 and
> Cifar100 and $r=6$ for Tiny-Imagenet to obtain a comparable number of trainable parameters. The training hyperparameters
> are the same as for AdaLoRA. As expected, LoRA is outperformed by both AdaLoRA and GeoLoRA, since the parameter budget
> is fixed per layer and cannot be used as effectively as in the adaptive methods.
>
> Lastly, we have added AdaLoRA to the stable diffusion benchmark, see Table 3, with scores between the ones reported for
> LoRA and GeoLoRa. We emphasize that both the best validation loss and the smallest parameter count with acceptable
> validation loss are achieved by the proposed GeoLoRA method.
>
> 3. **More Baselines**:
>
> As you mentioned, there are many LoRA-type methods, and in our manuscript, we tried to identify the best-performing
> methods from the current literature.
> We focused our comparisons on AdaLoRA, since this is the most popular rank-adaptive method. Since reporting more
> baselines was also a request of reviewer Ht32, we decided to add Lora+ and Dora comparisons since these methods have
> published code available. The results further highlight the advantages of GeoLoRA, and we thank the reviewer for pushing
> us to make an effort to compare against these further baselines. We refer to Table 1 for the corresponding results.
>
>
> Note: The Tables will be posted in subsequent comments.

---

> ### Author Response · Authors · 2024-11-16
> **Table 1: GLUE Benchmark Results including DORA, LoRA+, tuned AdaLoRA and mean values**
>
> We compare with full fine-tuning (Full FT), Houlsby adapter (HAdapter), Pfeiffer adapter (PAdapter), LoRA, DoRA, LoRA+, AdaLoRA, and BitFit. We report target metrics and computational performance (higher is better) for the median of 5 runs using different random seeds. Best results per dataset are shown in bold.
>
> We have tuned the global rank budget of AdaLoRa to match the parameter count of GeoLoRA and report the parameter counts of these two methods individually for each GLUE test case.
>
> | **Method** (_# Params_)            | **MNLI** (Acc) | **SST-2** (Acc) | **CoLA** (Mcc) | **QQP** (F1) | **QNLI** (Acc) | **RTE** (Acc) | **MRPC** (Acc) | **STS-B** (Corr) | **Mean** |
> |------------------------------------|----------------|------------------|----------------|--------------|----------------|---------------|----------------|------------------|----------|
> | **Full FT** (_184M_)               | 89.90          | 95.63           | 69.19          | 89.80        | 94.03          | 83.75         | 89.46          | 91.60           | 87.92    |
> | **BitFit** (_0.1M_)                | 89.37          | 94.84           | 66.96          | 84.95        | 92.24          | 78.70         | 87.75          | 91.35           | 85.77    |
> | **HAdapter** (_1.22M_)             | 90.13          | 95.53           | 68.64          | 89.27        | 94.11          | 84.48         | 89.95          | 91.48           | 87.94    |
> | **PAdapter** (_1.18M_)             | 90.33          | 95.61           | 68.77          | 89.40        | **94.29**      | 85.20         | 89.46          | 91.54           | 88.07    |
> | **LoRA r=8** (_1.33M_)             | 90.29          | 95.29           | 68.57          | 90.61        | 93.91          | 85.50         | 89.75          | 89.10           | 87.87    |
> | **LoRA+ r=8** (_1.33M_)            | *90.39*        | *95.37*         | *70.56*        | *90.65*      | *94.05*        | *85.59*       | *89.65*        | *88.07*         | *88.04*  |
> | **DoRA r=8** (_1.33M_)             | *90.11*        | *94.30*         | *68.50*        | **90.71**    | *94.31*        | *85.05*       | *89.32*        | *91.38*         | *87.96*  |
> | **AdaLoRA target r=8** (_1.27M_)   | **90.44**      | 95.64           | 68.76          | *90.65*      | 94.11          | **86.00**     | 89.44          | 91.41           | 88.30    |
> | **AdaLoRA (matched parameters)**   | *90.21* (_0.75M_) | 95.64 (_1.27M_) | 68.59 (_1.07M_) | *90.48* (_0.72M_) | 93.93 (_0.72M_) | 85.92 (_1.16M_) | 88.21 (_0.74M_) | 90.91 (_0.74M_) | *88.28* (_0.89M_) |
> | **GeoLoRA**                        | *90.38* (_0.7M_) | **95.98** (_1.17M_) | **69.03** (_0.98M_) | *90.53* (_0.69M_) | 94.23 (_0.70M_) | 85.93 (_1.19M_) | **90.10** (_0.75M_) | **91.58** (_0.71M_) | **88.47** (_0.86M_) |

---

> > ### Author Response · Authors · 2024-11-16
> > **Table 2: Vision Transformer including LoRA and Table 3: Stable Diffusion on Dreambooth including AdaLoRA**
> >
> > Table 2: Vit-base-patch16-224 fine-tuning on Cifar10, 100 and Tiny-Imagenet. We compare LoRa, AdaLoRA to GeoLoRA with local and global budgeting reporting the median of 5 runs using different random seeds. GeoLoRa "local" uses a layer-wise rank truncation, and "global" uses the same global rank budget criterion as AdaLoRA.
> >
> > | Method              | \# Params (Cifar 10) | Acc [%] (Cifar 10) | \# Params (Cifar 100) | Acc [%] (Cifar 100) | \# Params (Tiny-Imagenet) | Acc [%] (Tiny-Imagenet) |
> > |---------------------|----------------------|--------------------|-----------------------|---------------------|---------------------------|-------------------------|
> > | **LoRA**            | 0.47M (r=3)          | 98.47              | 0.47M (r=3)           | 91.47               | 0.99M (r=6)               | 87.34                   |
> > | **AdaLoRA**         | **0.47M**            | 98.51              | 0.45M                 | 91.44               | 0.90M                     | 87.21                   |
> > | **GeoLoRA, local**  | **0.47M**            | **98.55**          | **0.35M**             | **91.63**           | 0.92M                     | **88.09**               |
> > | **GeoLoRA, global** | 0.48M                | 98.51              | 0.47M                 | 91.62               | **0.75M**                 | 88.07                   |
> >
> >
> > Table 3: Stable Diffusion on Dreambooth benenchmark. We compare LoRA and GeoLoRA reporting the median of 5 runs. $r_0$ for AdaLoRA is the initial rank, while $r$ is the target rank.
> >
> > | Method                     | Val. Loss | \# Params |
> > |----------------------------|-----------|-----------|
> > | LoRA ($r = 5$)             | 0.275     | 3.0M      |
> > | LoRA ($r = 3$)             | 0.281     | 1.8M      |
> > | AdaLoRA ($r_0 = 8, r = 5$) | 0.245     | 4.7M      |
> > | AdaLoRA ($r_0 = 8, r = 3$) | 0.247     | 1.78M     |
> > | \ALGNAME{} ($\tau = 0.02$) | **0.242** | 2.6M      |
> > | \ALGNAME{} ($\tau = 0.1$)  | 0.257     | **1.4M**  |
> >
> >
> > We include all extended result tables in the revised manuscript.

---

> > > ### Comment · Reviewer_kJ1Q · 2024-11-26
> > >
> > > Thanks so much for your response and the new experiments. My major concern has be addressed. I will increase my score to 6.

---

### Author Response · Authors · 2024-11-16
**Updated Manuscript**

We thank the reviewers for taking their valuable time to review our submission and provide constructive feedback. We have revised the manuscript to account for the reviews, where we mark all revisions in *light blue*. In particular we

* Added LoRA+ and DoRA results for the GLUE benchmark
* Added parameter-matched AdaLoRA results, where we tune the target ranks of AdaLoRa so the corresponding parameter count approximately matches the parameter count of our proposed method, GeoLoRA. The results show that the proposed GeoLoRA method achieves higher scores at similar parameter levels compared to AdaLoRa.
* Added the average test score of all tested methods in the GLUE benchmark. In particular, the proposed GeoLoRA method has the highest average score.
* Added LoRA results for the Vision Transformer benchmarks
* Added AdaLoRa results for the Stable Diffusion benchmark

In both, the Vision Transformer and Stable Diffusion benchmark, LoRA scores lowest, GeoLoRA scores highest, and AdaLoRA scores are between the former two methods.

Lastly, we add in the introduction for each contribution where it can be found in the paper.

Kind regards,
Authors

---

### Comment · Area_Chair_Gjuz · 2024-11-24
**Reminder - Public Discussion Phase Ending Soon**

Dear PC memebers,

Thank you for your valuable comments during the review period, which raised many interesting and insightful questions. Now the discussion period is coming to a close, please take a moment to review the authors’ responses if you haven’t done so already. Even if you decide not to update your evaluation, kindly confirm that you have reviewed the responses and that they do not change your assessment.

Timeline: As a reminder, the review timeline is as follows:
November 26: Last day for reviewers to ask questions to authors.
November 27: Last day for authors to respond to reviewers.
November 28 - December 10: Reviewer and area chair discussion phase.
December 20: Meta reviews and initial decisions are due.


Thank you for your time and effort!

Best regards,
AC

---

### Meta-Review · Area_Chair_Gjuz · 2024-12-19

**Metareview:**

This paper introduces a new LoRA variant, GeoLoRA, designed for rank adaptivity, robustness, and computational efficiency. The reviewers universally acknowledged the motivation and novelty of the approach. Most concerns regarding the numerical experiments were effectively addressed by the authors during the rebuttal. The remaining issue, that GeoLoRA should be evaluated on larger datasets, is certainly important but does not, in my view, significantly impact the overall performance demonstrated by GeoLoRA. Therefore, I recommend acceptance.

**Additional Comments On Reviewer Discussion:**

The reviewers did not raise any fundamental concerns about GeoLoRA. Some detailed questions were effectively addressed during the rebuttal. Regarding the numerical experiments, reviewers had questions about the baseline, the improvement, and the model size. Most of these concerns were addressed by the authors, leading to an increase in scores. However, some issues related to larger models remain insufficiently discussed. Nevertheless, I believe this will not significantly impact the final decision.

---

### Decision · Program_Chairs · 2025-01-22

Accept (Poster)